# Geometric analysis enables biological insight from complex non-identifiable models using simple surrogates

**Alexander P. Browning**[1,2,3], **Matthew J. Simpson**[1,2]*

**1** School of Mathematical Sciences, Queensland University of Technology, Brisbane, Australia, **2** QUT Centre for Data Science, Queensland University of Technology, Brisbane, Australia, **3** Mathematical Institute, University of Oxford, Oxford, United Kingdom

* matthew.simpson@qut.edu.au

## Abstract

An enduring challenge in computational biology is to balance data quality and quantity with model complexity. Tools such as identifiability analysis and information criterion have been developed to harmonise this juxtaposition, yet cannot always resolve the mismatch between available data and the granularity required in mathematical models to answer important biological questions. Often, it is only simple phenomenological models, such as the logistic and Gompertz growth models, that are identifiable from standard experimental measurements. To draw insights from complex, non-identifiable models that incorporate key biological mechanisms of interest, we study the geometry of a map in parameter space from the complex model to a simple, identifiable, surrogate model. By studying how non-identifiable parameters in the complex model quantitatively relate to identifiable parameters in surrogate, we introduce and exploit a layer of interpretation between the set of non-identifiable parameters and the goodness-of-fit metric or likelihood studied in typical identifiability analysis. We demonstrate our approach by analysing a hierarchy of mathematical models for multicellular tumour spheroid growth experiments. Typical data from tumour spheroid experiments are limited and noisy, and corresponding mathematical models are very often made arbitrarily complex. Our geometric approach is able to predict non-identifiabilities, classify non-identifiable parameter spaces into identifiable parameter combinations that relate to features in the data characterised by parameters in a surrogate model, and overall provide additional biological insight from complex non-identifiable models.

## Author summary

Mathematical models play important roles in the interpretation of biological data. These models can be made arbitrarily complex, meaning issues related to parameter identifiability are relatively common. However, complex models with non-identifiable parameters can be useful to provide insight into the biological questions of interest, since they contain parameters of direct biological interest. In contrast, simpler identifiable models lack

**Data Availability Statement:** Code used to produce the results are available on Github at https://github.com/ap-browning/spheroid_geometry.

**Funding:** This work is funded by the Australian Research Council (https://www.arc.gov.au/) through the Discovery Project (DP200100177) awarded to MJS. The funders had no role in study design, data collection and analysis, decision to publish, or preparation of the manuscript.

biological granularity and comprise parameters that relate indirectly to the underlying biology through data features. In this work, we study the interrelationship between the non-identifiable parameters in a complex model and the identifiable parameters in a simple surrogate model. We aim to resolve the mismatch between model and data complexity by utilising the simple surrogate model to provide insight in cases where the parameters of interest cannot be determined from the available data. We demonstrate our approach by analysing mathematical models of multicellular tumour spheroid growth, an experimental model of cancerous tumour growth. Using the most fundamental and commonly reported measurements, we predict non-identifiabilities arising from different data collection regimes, and draw additional insight from complex models with non-identifiable parameters.

## Introduction

Mathematical models play an important role in the interpretation of data and the design of experiments. The complexity of many experiments and biological systems means that parameters relating to key biological mechanisms cannot be directly measured, but are rather quantified through the calibration of mechanistic mathematical models to experimental observations [1, 2]. Given that biological data are often limited and noisy, model parameters provide an objective means of quantifying observations and comparing behaviours across different types of experiments or different conditions within the same experiments [3, 4]. Minimising, or at least quantifying, parameter uncertainties is, therefore, of paramount importance for effective interpretation of experimental results.

A critical step in the application of mathematical models to interpret biological experiments is that of model selection [5–7]. Complex models—traditionally associated with a large number of unknown parameters—have potential to provide insights about a correspondingly large number of biological mechanisms, but often result in large parameter uncertainties when calibrated to typical experimental data [8–10]. Conversely, simpler models—including canonical models such as the logistic and Gompertz growth models—typically involve parameters that can be tightly constrained by data, but provide limited direct mechanistic insight [11].

In practice, model selection is routinely guided by information criterion; statistical metrics that quantify model parsimony, the trade-off between model fit and model complexity [7, 12]. One of many criteria used is the Akaike information criterion (AIC), given by

$$\text{AIC} = \underbrace{2k}_{\text{Complexity}} - \overbrace{2\ell(\hat{\mathbf{p}})}^{\text{Goodness-of-fit}}. \tag{1}$$

Here, $\hat{\mathbf{p}} \in \mathbb{R}^k$ is the maximum likelihood estimate (MLE), the $k$-dimensional parameter vector, $\mathbf{p}$, that produces the best model fit, and $\ell(\hat{\mathbf{p}})$ is the maximum log-likelihood, a measure of goodness-of-fit. In essence, AIC and other information criterion penalise complex models that produce marginally better goodness-of-fit over simpler models. Typically, AIC is computed for a range of candidate models that are ranked such that the model with the smallest AIC is the most favourable. To demonstrate, we consider the growth of multicellular tumour spheroids (Fig 1A), a complex, spatially heterogeneous biological system where often only simple

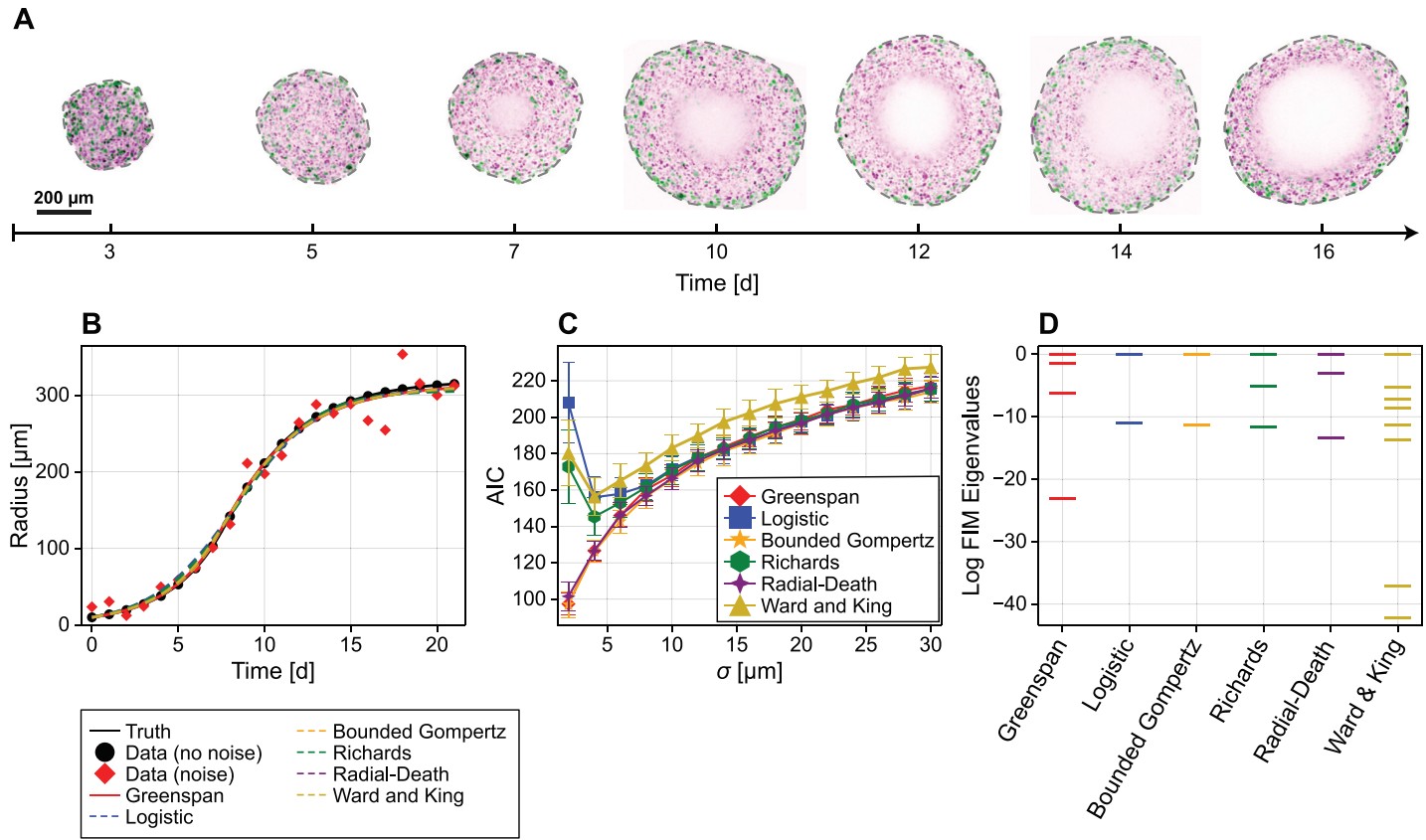

**Fig 1. Mathematical models of tumour spheroid growth.** (A) Microscopy images from tumour spheroid growth experiments. Spheroids are grown from WM983b cells (a human melanoma cell line) [18], harvested, and imaged using confocal microscopy at various time points. Cells are transduced with fluorescent cell cycle indicators, showing cells in gap 1 (purple) and gap 2 (green). From day 7, a necrotic core void of living cells is evident in the spheroid centre. (B) Synthetic spheroid data generated from Greenspan's model [13] (black discs) with additive normal noise with standard deviation $\sigma = 20$ μm (red diamonds). (B–C) Several mathematical models, including the Greenspan model, are able to match synthetic data. (C) AIC results for the model fitting exercise in (B) repeated over several values of the noise standard deviation. Shown is the mean and standard deviation from 100 repeats for each model. (D) Spectrum of the observed Fisher information matrix. Eigenvalues are shown on the log-scale and scaled such that the spectral radius is unity.

measurements, related to the overall size of radius of spheroids, are typically available throughout an experiment. We generate synthetic radius measurements from a mathematical model of intermediate complexity (the Greenspan model with $k = 4$ parameters) that was recently validated against experimental data for the first time [13, 14]. We corrupt measurements with normally distributed measurement noise with standard deviation $\sigma$ and attempt to distinguish between a range of spheroid growth models, with complexity ranging from the logistic growth model ($k = 2$) [15] to the complex multiphase spatial model of Ward and King ($k = 8$) [16, 17]. In Fig 1B we set $\sigma = 20$ μm and in Fig 1C we vary $\sigma$. Once calibrated, all models lead to predictions of the spheroid radius that are visually indistinguishable (Fig 1B), and all except for Ward and King's model are indistinguishable using AIC for a sufficiently large, and biologically realistic, noise standard deviation (Fig 1C). Full mathematical details of all models are given in *Models*.

Aside from being unable to distinguish between models in the tumour spheroid example, criterion-based choices cannot account for the biological question—or more specifically, the biological mechanisms—of interest. For example, the logistic growth and Gompertz growth models produce an excellent match to synthetic tumour spheroid data and quantify behaviour in terms of a growth rate parameter and long-time limiting spheroid size. However, these

models cannot provide information relating to the mechanisms that govern growth or determine the long-time limiting spheroid size; mechanisms such as sensitivity to and availability of oxygen and other essential nutrients. More recently, the mathematical modelling literature has moved toward tools such as parameter identifiability analysis to guide model selection [19–21]. Identifiability analysis can determine if model parameters are identifiable and can be estimated from data; both in a theoretical noise-free data limit (*structural* identifiability) [22–25], and in the more realistic case of a finite amount of noisy data (*practical* identifiability) [19, 26]. In comparison to model selection criterion like AIC, identifiability analysis provides information about the identifiability of individual model parameters. While a complex model may have a large number of non-identifiable parameters and a high AIC value, it may still prove useful provided the parameters of interest (for example, the oxygen sensitivity) are identifiable.

In the vicinity of the MLE, the identifiability of model parameters can be assessed using the local curvature of the expected log-likelihood function, also known as the Fisher information matrix (FIM), denoted $I(\hat{\mathbf{p}}) \in \mathbb{R}^{k \times k}$. The FIM is a $k \times k$ positive semi-definite matrix that quantifies the amount of information about the parameters contained in the data, and has both a statistical and geometric interpretation. Statistically, the inverse of the FIM provides a lower-bound on the covariance of parameter estimates. Therefore, a FIM that is singular corresponds to at least one model parameter that can only be estimated with infinite variance and, therefore, cannot be determined from data. Geometrically, the FIM is related to the Hessian of the log-likelihood function and therefore contains information about the directions in parameter space in which the log-likelihood (and therefore the model) is sensitive and directions in which the log-likelihood is insensitive [27]. Specifically, the eigenvalues of the FIM correspond to the curvature in the direction of the corresponding eigenvectors; eigenvectors associated with zero or near-zero eigenvalues correspond to directions in parameter space (also referred to as eigenparameters) to which the model output is insensitive [28, 29]. Conversely, eigenvectors associated with relatively large eigenvalues give informative directions; the directions to which the model is most sensitive. So-called analysis of *model sloppiness* is concerned with studying the spectrum of the FIM to determine the number of *sloppy*, or insensitive, eigenparameters in a model [8, 30–32]. To demonstrate, in Fig 1D we show the spectrum of the FIM for each tumour spheroid model. As the relative difference between eigenvalues is scale-dependent, it is difficult to interpret results from the two parameter models. However, results for the Greenspan and Ward and King models show two disparate clusters of eigenvalues, indicating a group of informative directions (corresponding to eigenvalues that are relatively large), and a group of uninformative or sloppy directions (corresponding to eigenvalues that are closer to zero).

For the Greenspan model, the single insensitive direction identified from analysis of model sloppiness corresponds to a one-dimensional manifold (i.e., a curve) in parameter space along which the parameters can be identified. At the core of identifiability and sloppiness analysis is that data are unable to constrain the model parameter space to a point estimate, but rather a one- or higher-dimensional manifold [33]. Of practical application, analysis of this manifold allows for model reduction, where the number of parameters in a model can be reduced by pre-constraining or removing sloppy eigenparameters without significantly reducing the predictive power of a model [34, 35]. However, to date, analysis of the interrelationship between models using the model parameter manifold has been constrained to simpler models *nested* within a complex model; that is, where simpler models can be recovered by placing constraints on the parameters in the complex model, for example by setting certain parameters to zero. Examples of nested models include recovering the logistic growth model from the Fisher-Kolmogorov model by assuming the population is well mixed [36], and recovering the Gompertz

or logistic growth models from Richards growth model by constraining the shape parameter [21]. Moreover, the FIM is based on the expected log-likelihood, a one-dimensional measure of overall model fit that determines manifolds in parameter space to which parameters are constrained by data or to which the model output is insensitive. FIM-based tools cannot, therefore, provide information about how *features* of the model output change with parameters.

Our contribution is to study models with non-identifiable parameters using identifiable models that produce quantitatively similar behaviour; models that may be indistinguishable from information-criterion based analysis. To study the interrelationship between parameters in any two models (nested or non-nested), we define *model equivalence* in the least-squares sense, and study the associated map from the parameters in a complicated, possibly heavily-parameterised and non-identifiable model, to parameters in a simpler, identifiable model (Fig 2B). For example, we study identifiability of mechanistic ordinary and partial differential equation (ODE and PDE) models of tumour spheroid growth—relatively complicated models containing parameters quantifying nutrient sensitivities, oxygen diffusion, and oxygen consumption—through simple models like the well known logistic and Gompertz growth models that do not explicitly incorporate biophysical mechanisms that influence growth, but rather describe behaviour with largely phenomenological parameters such as the early-time growth rate and long-time limiting spheroid size.

We demonstrate our framework through identifiability analysis of tumour spheroid data. Noisy measurements relating to the outer radius of tumour spheroids are collected (Fig 1A) and quantified with models ranging from the phenomenological logistic growth model, to detailed spatial models involving coupled nonlinear PDEs which require experimental measurements in addition to spheroid radius to parameterise [14]. We work with synthetic data generated from a model of intermediate complexity, the Greenspan model (Fig 1B), and present a series of new and existing models from the literature that produce similar agreement with the data. Initially, we focus on models with a small number of parameters so that model

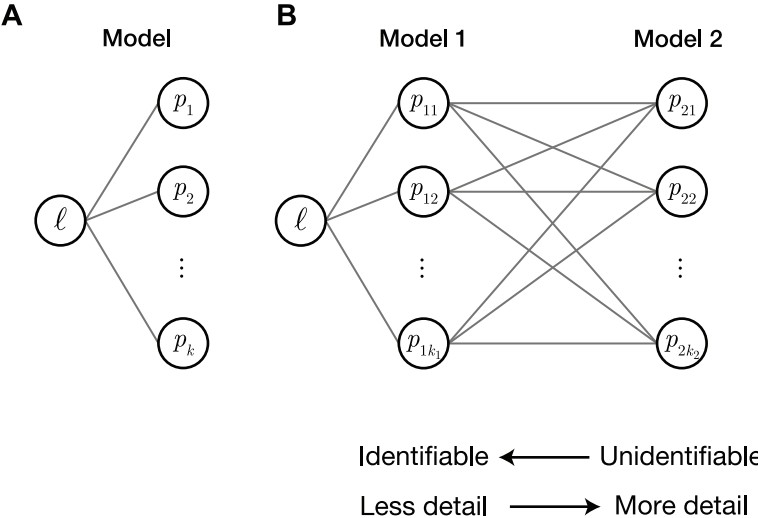

**Fig 2. Studying identifiability through between-model geometry.** (A) Typically, model parameters, $\mathbf{p} = [p_1, p_2, ..., p_k]^\top$ are considered functions of the log-likelihood, $\ell(\mathbf{p})$, a one-dimensional metric of model fit. Non-identifiability of model parameters is characterised by insensitivity of the likelihood to a parameter or a parameter combination. (B) We consider a range of models, each parameterised by $\mathbf{p}_i = [p_{i1}, p_{i2}, ..., p_{ik_i}]^\top$. We then study the functional relationships between parameters of different models (grey lines).

equivalence manifolds can be visualised in $\mathbb{R}^3$. Subsequently, we study Ward and King's model, a model with a large number of parameters for which we must rely on non-graphical means, such as the sensitivity matrix and the Jacobian of the model link, for analysis. Aside from the requirement that the Jacobian of model outputs with respect to parameters be available, which may limit our analysis to primarily to models that are deterministic, we expect our methodology to generalise to any hierarchy of models in biology and systems biology.

## Models

We study a hierarchy of mathematical models that describe the time-evolution of tumour spheroid radius. Such models have a long history in the mathematical biology literature, and range from simple sigmoid growth models [11], to models that describe the spatial distribution of cells in spheroids and the eventual saturation of growth due to nutrient deprivation and mass loss due to necrosis in the tumour core [17]. We generate synthetic data using the canonical Greenspan model [13][14, 18]. Therefore, we treat the Greenspan model as the *true* model, and the corresponding set of parameters as the true parameters. In this section, we present the mathematical models we use for analysis.

For all models, we denote the spheroid radius by $R(t)$, and fix the initial spheroid size $R(0)$ = $R_0$ = 10 μm as a known parameter that can be directly measured from data. The choice of $R_0$ = 10 μm $\ll R_{max}$ is made to ensure that the simple models we consider are identifiable. We consider that data comprise of spheroid radius measurements at discrete observation times $T = [t^{(1)}, t^{(2)}, ..., t^{(n)}]^{\mathsf{T}}$ and denote predictions from model $i$ by

$$\mathbf{m}_i(\mathbf{p}) = [R_i(t^{(1)}; \mathbf{p}), R_i(t^{(2)}; \mathbf{p}), ..., R_i(t^{(n)}; \mathbf{p})]^{\mathsf{T}}. \tag{2}$$

The formulation in Eq (2) is highly flexible: for example, we could incorporate model predictions relating to the inner structure of the spheroid by appending elements to the end of $\mathbf{m}_i(\mathbf{p})$. In this work, we set $T = [0, 1, 2, ..., 21]^{\mathsf{T}}$ d, as shown in Fig 1B, and denote $R_i(t; \mathbf{p})$ predictions of the spheroid radius at time $t$ from model $i$. In this work we take great care to connect our mathematical models with experimental data by working with a combination of dimensional and non-dimensional quantities. We are motivated by experimental data of tumour spheroids that comprise observations of spheroid radius over a period of approximately 16 days [14, 18]. However, standard imaging techniques do not provide measurements of cell densities or nutrient concentrations. Therefore, we non-dimensionalise dependent variables relating to cell densities and nutrient concentrations, and leave the independent variables related to measurable time and space scales dimensional.

### Model 1. Greenspan's model ($k$ = 4)

First, we consider the canonical Greenspan model [13] that describes spherically symmetric spheroid growth due to cell proliferation dependent on a nutrient (such as oxygen) that diffuses into the spheroid from the surrounding medium (Fig 3A). We have previously validated the Greenspan model against experimental data [14]. Spheroid growth progresses through the three phases observed in experimental data (Fig 1A; [18, 37, 38]).

First, if the initial spheroid size is sufficiently small, spheroids progress through an exponential growth phase, where nutrient is available throughout the spheroid above the minimum threshold concentration required for cell proliferation. We denote the critical concentration as $c_1 = \omega_1/\omega_\infty$, where $\omega_1$ mol μm$^{-3}$ is the threshold concentration required for cell proliferation, and $\omega_\infty$ mol μm$^{-3}$ is the concentration in the surrounding medium and at the spheroid boundary. During this first phase, cells proliferate exponentially at a per-unit-volume rate of $\lambda$ d$^{-1}$.

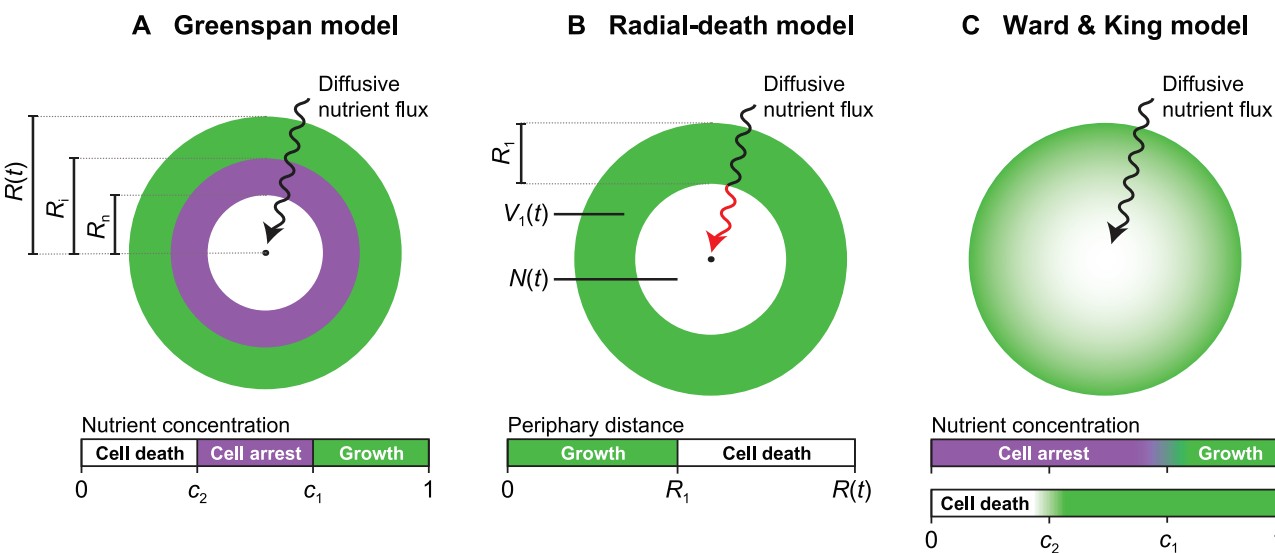

**Fig 3. Schematic of the spatial spheroid models considered.** (A) The Greenspan model describes nutrient-limited growth, where cell proliferation is dependent upon the relative availability of a diffusive nutrient. Cells proliferate in regions of the spheroid where the nutrient concentration is sufficiently high; enter cell-cycle arrest, but do not die, in regions where the nutrient concentration is too low for growth, but above the threshold for life; and die in regions where the nutrient concentration is sufficiently low. (B) The radial-death model describes an implicitly modelled diffusive nutrient, which is assumed to drop below the threshold required for cell proliferation at a constant distance from the spheroid periphery. (C) Ward and King's model describes both cell proliferation and death as dependent on a diffusive nutrient that is explicitly modelled as a function of space.

Fluorescent cell cycle indicators indicate that spheroids eventually enter a second phase of growth where cell proliferation in the centre of the spheroid is inhibited such that cells remain viable but enter cell cycle arrest. We make the standard assumption that this is due to the nutrient concentration falling below the relative concentration $c_1$ at the spheroid centre, but remaining above the relative concentration required for cell viability, denoted by $c_2$ (Fig 3A). During this second phase, cells located close to the spheroid periphery remain proliferative since the nutrient density is sufficiently high in this region.

Finally, spheroids progress to a size such that the nutrient concentration at the centre of the spheroid is lower than that required for viability. Cells in the centre of spheroids die, leading to the formation of a necrotic core and resulting in a per-unit-volume mass loss of $\zeta$ d$^{-1}$. This phase is evident in experimental measurements from day 7 (Fig 1A). In summary, the Greenspan model predicts the eventual formation of a three-layered compound sphere with a central necrotic core, an intermediate shell of living but non-proliferative cells, and an outer shell of living proliferative cells (Fig 3A). This final structure is consistent with experimental observations of spheroid growth shown in Fig 1A [18].

While the Greenspan model explicitly incorporates a proliferation and death process dependent upon a spatially diffusive nutrient, the assumption that the nutrient-related dependencies are Heaviside functions yields a series of implicit analytical expressions for the radius of the inhibited region, $R_i(t)$, and radius of the necrotic region, $R_n(t)$, in terms of the overall spheroid radius (Fig 3A) [13]. We define three composite parameters

$$Q = \sqrt{\frac{1 - c_1}{1 - c_2}} \in (0,1), \qquad R_d = \sqrt{\frac{6D_c(\omega_\infty - \omega_2)}{\alpha}} > 0, \qquad \gamma = \frac{\zeta}{\lambda} > 0, \qquad (3)$$

where $\alpha$ mol d$^{-1}$ and $D_c$ μm$^2$ d$^{-1}$ is the nutrient consumption rate and diffusivity, respectively.

Therefore, $R_i(t)$ and $R_n(t)$ are given by the solution of the following algebraic equations [13]

$$R_n(t) = 0,$$
$$R_i(t) = \sqrt{R^2(t) - Q^2 R_d^2}, \tag{4}$$

and

$$0 = R^3(t) - R_d^2 R(t) - 3R(t)R_n^2(t) - 2R_n^3(t),$$
$$0 = Q^2 R_d^2 R(t)R_i(t) + R(t)R_i^3(t) + 2R(t)R_n^3(t) - R^3(t)R_i(t) - 2R_i(t)R_n^3(t), \tag{5}$$

during phases two and three, respectively. During the first phase, $R_i = R_n = 0$. Greenspan [13] also showed that the first phase applies for $R(t) < QR_d$, the second for $QR_d \leq R(t) < R_d$ and the third for $R(t) \geq R_d$.

Overall, the time-evolution of the spheroid radius is governed by the ODE

$$\frac{dR(t)}{dt} = \frac{\lambda}{3} R(t) \left( 1 - \frac{R_i^3(t)}{R^3(t)} - \gamma \frac{R_n^3(t)}{R^3(t)} \right). \tag{6}$$

Here, we have expressed $dR(t)/dt$ in the form of a generalised logistic growth model

$$\frac{dR(t)}{dt} = \frac{\lambda}{3} R(t) f(R(t)), \tag{7}$$

where $\lambda$ is the *volumetric* growth rate for $f(R(t)) = 1$ (the factor of 1/3 arises from applying the chain rule to convert from working in terms of spheroid volume to spheroid radius). $f(R)$ is sometimes referred to as a *crowding function*, defined such that $f(R) \rightarrow 0$ for $R$ sufficiently large. We show the solution to the Greenspan model and the corresponding crowding function in Fig 4 using parameters given in Table 1. The Greenspan model depends on four unknown parameters $\mathbf{p}_1 = [Q, R_d, \gamma, \lambda]^\top$.

## Model 2. Logistic model ($k = 2$)

The logistic model is used widely throughout biology and ecology as, for example, a model of *in vitro* cell growth [11, 15, 39–41] and coral regrowth [21, 41]. Whereas in the Greenspan

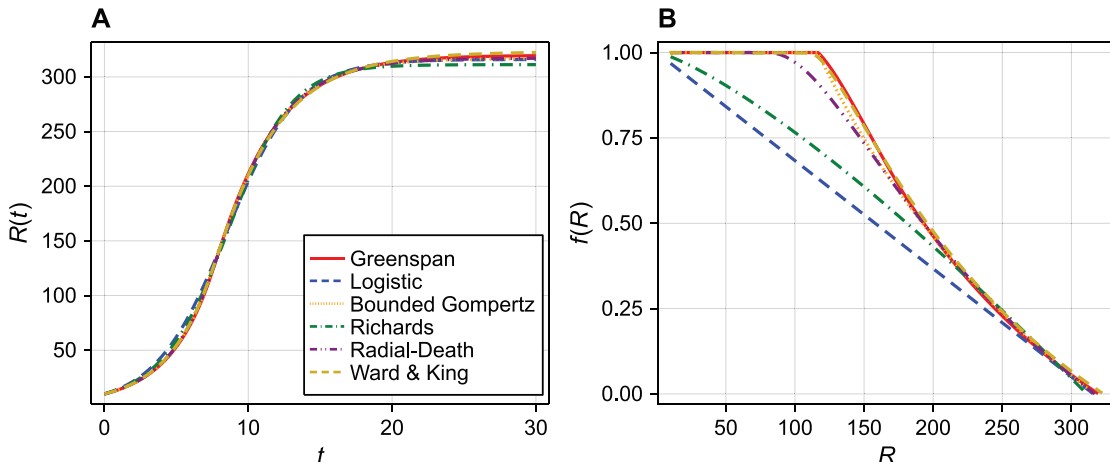

**Fig 4. Model solutions and crowding functions.** (A) We show the outer radius, $R(t)$, predicted by each model using parameters chosen to match the solution of the Greenspan model. (B) The corresponding crowding function for each model when expressed as a generalised logistic growth equation (Eq (7)).

**Table 1. Summary of model parameters.** Synthetic data are generated using the Greenspan model, with parameters $\lambda$, $R_d$, $Q$, and $\gamma$. Parameters in all other models are chosen as the parameter set that gives the closest match to synthetic data from the Greenspan model.

| Parameter | Description | Value | Units | Models |
|---|---|---|---|---|
| $R_0$ | Initial spheroid radius. | 10 | μm | [1]–[6] |
| $\lambda$ | Maximum volumetric growth rate. | 1 | $d^{-1}$ | [1]–[6] |
| $R_d$ | Spheroid radius before necrosis. | 150 | μm | [1], [5] |
| $Q$ | Eq (3). | 0.8 | – | [1] |
| $\gamma$ | $\zeta/\lambda$. | 1 | – | [1] |
| $R_{max}$ | Maximum spheroid radius. | – | μm | [2]–[4] |
| $\beta$ | Shape parameter in Richards model. | – | – | [4] |
| $\zeta$ | Necrotic material loss rate. | – | $d^{-1}$ | [5] |
| $\delta$ | Maximum volumetric death rate. | – | $d^{-1}$ | [5] |
| $c_1$ | Relative nutrient concentration for median growth. | – | – | [6] |
| $c_2$ | Relative nutrient concentration for median death. | – | – | [6] |
| $\alpha$ | Relative nutrient consumption rate. | – | – | [6] |
| $D_p$ | Cellular material diffusivity. | – | $\mu m^2\, d^{-1}$ | [6] |
| $Q_p$ | Rate of cellular material inflow at spheroid boundary. | – | $\mu m\, d^{-1}$ | [6] |
| $p_0$ | Cellular material concentration in medium. | – | – | [6] |
| $n_0$ | Initial spheroid density. | – | – | [6] |

model assumptions relating cell proliferation to the local density of a diffusive nutrient yield a crowding function that eventually caused overall growth to cease, in the logistic model we make the simplistic assumption that spheroid growth eventually ceases when the size reaches a maximum radius, $R_{max}$ μm [42]. The logistic model and generalisations thereof (including the Gompertz and Richards models [42]) are phenomenological in the context of tumour spheroid models; they are not explicitly constructed from biological mechanisms by which overall growth is inhibited and eventually ceases.

While the logistic growth model is commonly used to describe the time-evolution of spheroid *volume* [11], we find that the time-evolution of spheroid *radius* in our synthetic data is more consistent with logistic growth (Fig 1B). The logistic model for spheroid radius is given by

$$\frac{dR(t)}{dt} = \frac{\lambda}{3}\, R(t) \left( 1 - \frac{R(t)}{R_{max}} \right). \tag{8}$$

Here, the factor of 1/3 arises from working in terms of spheroid radius instead of spheroid volume. Both the Greenspan and logistic models describe exponential growth at the per-volume rate of $\lambda\, d^{-1}$ for sufficiently small spheroids, $R(t)/R_{max} \ll 1$. However, the logistic model differs in that it does not include an initial period of time where growth is exponential. We demonstrate this by comparing the crowding function for the logistic model to that of the Greenspan model in Fig 4; in the logistic model, $f(R)$ is a linearly decreasing function of $R$. Therefore, in the logistic model, the overall growth rate of the spheroid is never equal to the maximum growth rate of $\lambda/3$. Rather, it is always less than this maximum for $R(t) > 0$. The logistic model depends on two unknown parameters $\mathbf{p}_2 = [\lambda, R_{max}]^\intercal$.

## Model 3. Bounded Gompertz model ($k = 2$)

Gompertz's growth model has been used since the 1960s to describe the growth of solid tumours [11, 43, 44], and is given by

$$\frac{\mathrm{d}R(t)}{\mathrm{d}t} = \frac{\lambda}{3} \, R(t)\log\!\left(\frac{R_{\max}}{R(t)}\right). \tag{9}$$

One feature of the Gompertz model is that the growth rate is unbounded for $R(t)/R_{\max} \ll 1$. As we do not see this in the experiments or the Greenspan model, we consider a *bounded Gompertz* model by setting

$$\frac{\mathrm{d}R(t)}{\mathrm{d}t} = \frac{\lambda}{3} \, R(t)\min\!\left(\log\!\left(\frac{R_{\max}}{R(t)}\right), 1\right). \tag{10}$$

Therefore, the bounded Gompertz model undergoes a period of exponential growth for $R(t) < R_{\max}/\exp(1)$ before growth becomes inhibited, which is similar to the first phase of growth described by the Greenspan model. The solution of the bounded Gompertz model and the corresponding crowding function is shown in Fig 4. The bounded Gompertz model depends on two unknown parameters $\mathbf{p}_3 = [\lambda, R_{\max}]^{\top}$ each with the same interpretation as those in the logistic model.

## Model 4. Richards' model ($k = 3$)

Richards' model interpolates between the logistic and standard Gompertz models by introducing an additional shape parameter into the crowding function, $\beta$, that alters the shape of the solution [21, 42, 45]. The Richards model is given by

$$\frac{\mathrm{d}R(t)}{\mathrm{d}t} = \frac{\lambda}{3}R(t)\left(1 - \left(\frac{R(t)}{R_{\max}}\right)^{\beta}\right). \tag{11}$$

The logistic model can be recovered by setting $\beta = 1$, and the standard formulation of the Gompertz model in the limit $\beta \to 0^{+}$. The solution to the Richards model and the corresponding crowding function is shown in Fig 4. The Richards model has three unknown parameters $\mathbf{p}_4 = [\lambda, R_{\max}, \beta]^{\top}$.

## Model 5. Radial-death model ($k = 3$)

We introduce the *radial-death model*, a simplistic compartment model that captures the key elements of Greenspan's model, namely that the inhibition of overall growth is caused by the nutrient deprivation in the spheroid core. Due to consumption by cells, the nutrient concentration is a decreasing function of the distance to the spheroid periphery, so we assume that nutrients (such as oxygen) can diffuse into the spheroid up to a distance $R_{\mathrm{d}}$ μm from the spheroid periphery before reaching a critically low concentration where cell proliferation ceases and cell death begins (Fig 3B). While $R(t) \leq R_{\mathrm{d}}$, the spheroid is composed proliferative cells, and for $R(t) > R_{\mathrm{d}}$ of an expanding annulus of constant density with thickness $R_{\mathrm{d}}$, and a necrotic core of radius $R(t) - R_{\mathrm{d}}$. We assume that living cells proliferate at per-volume rate $\lambda \, \mathrm{d}^{-1}$, and necrotic material is lost at per-volume rate $\zeta \, \mathrm{d}^{-1}$.

Denoting the volume of living and necrotic cells $V_1(t)$ and $N(t)$, respectively, the dynamics are governed by

$$
\begin{aligned}
\frac{\mathrm{d}V_1(t)}{\mathrm{d}t} &= \lambda V_1(t) - g(V_1(t)), \\
\frac{\mathrm{d}N(t)}{\mathrm{d}t} &= g(V_1(t)) - \zeta N(t).
\end{aligned}
\tag{12}
$$

Here, $g(V_1(t))$ represents the transfer of living cells in the periphery annulus to the necrotic core at the spheroid centre to maintain an annulus width less than $R_d$ (Fig 3B). Note that $g(V_1(t)) = 0$ for $R(t) < R_d$. Denoting the total spheroid volume $V(t) = V_1(t) + N(t) = 4\pi R^3(t)/3$, we see that

$$
\frac{\mathrm{d}V(t)}{\mathrm{d}t} = \lambda V_1(t) - \zeta N(t),
\tag{13}
$$

or equivalently

$$
\frac{\mathrm{d}R(t)}{\mathrm{d}t} = \frac{\lambda}{3}\, R(t)\left[1 - \max\left(0, \frac{(R(t) - R_d)^3}{R^3(t)} - \frac{\zeta}{\lambda}\frac{(R(t) - R_d)^3}{R^3(t)}\right)\right].
\tag{14}
$$

Therefore, the radial-death model is fully described by a single independent variable $R(t)$ (or equivalently $V(t)$).

We show the solution to the radial-death model and the corresponding crowding function in Fig 4. The radial-death model has three unknown parameters, $\mathbf{p}_5 = [\lambda, \zeta, R_d]^{\mathsf{T}}$. All three parameters have an equivalent interpretation in the Greenspan model, and $\lambda$ has an equivalent interpretation in all other models.

## Model 6. Ward and King's multiphase model ($k = 8$)

Lastly, we consider the growth-saturation spheroid model of Ward and King [16, 17], a moving boundary PDE model that explicitly incorporates the cell density, cellular material density (i.e., the DNA, lipids, proteins, and other material that living cells are composed of), and nutrient density, as functions of space and time (Fig 3C). By assuming that spheroid growth is spherically symmetric, we end up working with a system of three time-dependent PDEs with one spatial coordinate. Full details are available in [17], however we apply several simplifications and so now provide a summary of the key mechanics in the model.

We denote the relative density of living cells $n(r, t)$, that of cellular material $p(r, t)$, and that of nutrient (i.e., oxygen) $c(r, t)$. Here, $0 \leq r \leq R(t)$ is the spatial variable describing the distance from the spheroid centre to the moving spheroid boundary at $r = R(t)$. We assume that the spheroid contains no voids so that $1 = n(r, t) + p(r, t)$, and that living cells and cellular material are incompressible and are transported throughout the spheroid with velocity $v(r, t)$. Cells are subject to a maximum per-unit-volume growth rate of $\lambda\ \mathrm{d}^{-1}$, which is a increasing function of nutrient concentration, specified by the Hill function,

$$
k_m(C) = \frac{\lambda c^{m_1}}{c_1^{m_1} + c^{m_1}}.
\tag{15}
$$

We fix the Hill exponent $m_1 = 10$ to capture the Heaviside-like switch behaviour in Greenspan's model, and $c_1$ is the nutrient concentration at which the proliferation rate is half of the maximum. Similarly, cells are subject to the nutrient-dependent death rate, where the death

rate is a decreasing function of nutrient concentration,

$$k_d(C) = \delta\left(1 - \frac{c^{m_2}}{c_2^{m_2} + c^{m_2}}\right).$$ (16)

Again, we fix $m_2 = 10$ and $c_2$ is the nutrient concentration for which the death rate is half of the maximum rate, denoted by $\delta$ d$^{-1}$. We assume that nutrient is consumed at rate $k(C) = \alpha k_m(C)$. Cellular material, $p(r, t)$ is assumed to have the same density as living cells, is consumed during mitosis, diffuses freely throughout the spheroid with diffusivity $D_p$ μm$^2$ d$^{-1}$, is available in the surrounding media at relative density $p_0$, and enters the spheroid from the surrounding media at the spheroid boundary at flux $Q_p$ μm d$^{-1}$.

Since the nutrients diffuse faster that cells proliferate, we assume that the nutrient is in diffusive equilibrium. These assumptions give rise to the coupled system of PDEs

$$
\begin{aligned}
\frac{\partial n}{\partial t} + v\frac{\partial n}{\partial r} &= n\frac{D_p}{r^2}\frac{\partial}{\partial r}\left(r^2\frac{\partial n}{\partial r}\right) + (k_m(C) - k_d(C))n, && t > 0, && 0 < r < R(t), \\
0 &= \frac{1}{r^2}\frac{\partial}{\partial r}\left(r^2\frac{\partial c}{\partial r}\right) - k(C)n, && t > 0, && 0 < r < R(t), \\
0 &= \frac{D_p}{r^2}\frac{\partial}{\partial r}\left(r^2\frac{\partial n}{\partial r}\right) + \frac{1}{r^2}\frac{\partial(r^2 v)}{\partial r} && t > 0, && 0 < r < R(t), \\
n(r, t) &= n_0, \quad r(t) = R_0, && t = 0, && 0 < r < R(t), \\
\frac{\partial n}{\partial r} &= 0, \quad \frac{\partial c}{\partial r} = 0, \quad v = 0, && t > 0, && r = 0, \\
c = 1, \quad -D_p\frac{\partial n}{\partial r} &= -Q_p(1 - p_0 - n), \quad \frac{dR(t)}{dt} = v, && t > 0, && r = R(t).
\end{aligned}
$$ (17)

The moving boundary $r = R(t)$ corresponds to the outer radius of the spheroid. Here, $v(r, t)$ denotes the velocity of material throughout the spheroid, describes the movement of both cells and cellular material. Therefore, the velocity at the spheroid boundary, $v(R(t), t)$, corresponds to the growth rate of the spheroid outer radius. In contrast to the other models considered in this work, overall spheroid growth in the Ward and King model is driven explicitly by the spatial structure and composition of material inside the spheroid.

While the Ward and King model cannot be expressed in the form of Eq 7 generalised logistic as a generalised logistic growth model, we can calculate a crowding function empirically

$$f(R) = \frac{3}{\lambda R(t)}\frac{dR}{dt}.$$ (18)

In Fig 4, we show the solution to the Ward and King model and the corresponding empirical crowding function. Details of the numerical algorithm used to solve the Ward and King model are given as supplementary material (S1 File).

The Ward and King model has eight unknown parameters, $\mathbf{p}_6 = [\lambda, \delta, c_1, c_2, \alpha, D_p, Q_p, p_0]^\top$. Only the per-unit-volume proliferation rate, $\lambda$, shares an interpretation with all of the other models. The remaining seven parameters relate to the mechanics of spheroid growth.

## Results

There are two main components to our analysis. First, we apply standard approaches to determine the practical identifiability of the Greenspan model and two simplistic models in one

hierarchy using synthetically generated, noisy data. Second, we develop our novel geometric approach to study non-identifiabilities in the Greenspan model using the geometric relationship between models in this hierarchy. As this geometric analysis considers features of the model outputs, which are deterministic and do not depend on data, this analysis is akin to both structural identifiability and sensitivity analysis. However, as the models are not nested, model outputs do not become identical in the no-noise limit, and so our analysis implicitly incorporates modelling bias which is a feature of analysis of most experimental data, where every mathematical model is an abstraction.

## Identifiability analysis

We make the standard assumption that observations, denoted $\mathbf{y} = [y^{(1)}, y^{(2)}, \ldots, y^{(n)}]^{\mathsf{T}}$, are subject to independent additive normal noise [46, 47], such that

$$y^{(k)} = \mathbf{m}_i^{(k)}(\mathbf{p}_i) + \varepsilon \tag{19}$$

where $\varepsilon \sim N(0, \sigma^2)$ and $\mathbf{m}_i^{(k)}(\cdot)$ denotes the $k$th element of $\mathbf{m}_i(\cdot)$. In our case, $\mathbf{m}_i(\mathbf{p}) = R(t^{(k)}; \mathbf{p})$. Therefore, the log-likelihood function is given by

$$\ell_i(\mathbf{p}_i) = -\log(\sigma\sqrt{2\pi}) - \frac{1}{2\sigma^2} \|\mathbf{y} - \mathbf{m}_i(\mathbf{p}_i)\|^2. \tag{20}$$

The *maximum likelihood estimate* (MLE) is the parameter vector that maximises the log-likelihood function, or equivalently, minimises the error term or loss function

$$e_i(\mathbf{p}_i) = \|\mathbf{y} - \mathbf{m}_i(\mathbf{p}_i)\|^2 = \sum_{k=1}^{n} \left( \mathbf{y}^{(k)} - \mathbf{m}_i^{(k)}(\mathbf{p}_i) \right)^2. \tag{21}$$

Therefore, the MLE is equivalent to the least-squares estimate. We calculate the MLE for each model using synthetic data generated from the Greenspan model with a pre-specified constant noise with standard deviation $\sigma = 20$ μm in Fig 1B, demonstrating that all models are capable of producing an excellent fit to the synthetic data.

**Profile likelihood.**   To establish the identifiability of individual model parameters, we apply the profile likelihood method [19, 48, 49]. The profile likelihood method *profiles* the log-likelihood function by finding the MLE subject to the constraint that the profiled parameter is fixed. Denoting the parameter to be profiled by $\varphi$ and the remaining parameters as $\boldsymbol{\eta}$ such that $\mathbf{p} = (\varphi, \boldsymbol{\eta})$, the profile log-likelihood is given by

$$\mathrm{PLL}(\varphi) = \sup_{\boldsymbol{\eta}} \ell_i(\varphi, \boldsymbol{\eta}) - \ell_i(\hat{\mathbf{p}}_i), \tag{22}$$

where $\hat{\mathbf{p}}_i$ is the MLE when all parameters are varied. The value of the profile likelihood at $\varphi = \varphi_0$ corresponds to the test-statistic for the likelihood ratio test and has an asymptotic $\chi^2$-distribution. Therefore, we can establish identifiability by comparing the profile log-likelihood of a parameter to the threshold for an approximate 95% confidence interval, equal to $-\Delta_{1,0.95}/2 \approx -1.92$, where $\Delta_{\nu,q}$ refers to the $q$th quantile of a $\chi^2$ distribution with $\nu$ degrees of freedom [50]. Model parameters with tightly-constrained intervals for which the profile likelihood is above this threshold are classified as identifiable. In this work, we take the supremum of the log-likelihood function numerically using the Nelder-Mead algorithm over a region that covers the true parameters over several orders of magnitude [51]. We profile the log-likelihood in sequence, starting at the point closest to the MLE. To ensure we find the global maximum, we initiate the optimisation routine at three points, corresponding to the MLE, the previously

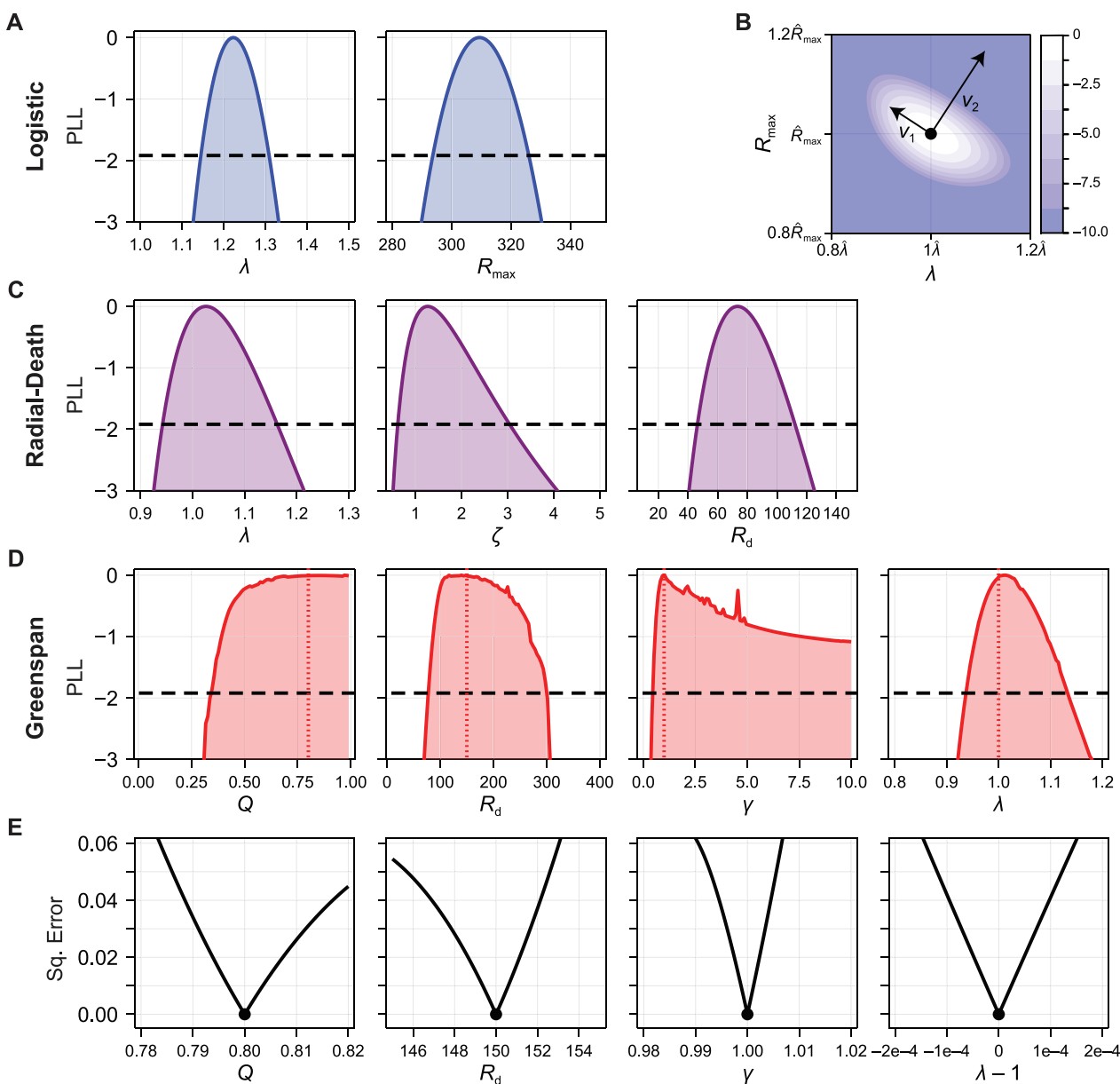

**Fig 5. Identifiability analysis for the logistic, radial-death, and Greenspan models.** (A,C,E) Profile likelihood for the parameters in each model, using synthetic data generated from the Greenspan model with standard deviation σ = 20 μm. Also shown is the threshold for an approximate 95% confidence interval (black-dashed). Decreasing the confidence level (i.e. from 95% to 99%) lowers this threshold. For the Greenspan model, we additionally show the parameter values used to generate the synthetic data. (E) Shows the profiled error function for each parameter in the Greenspan model. Note that profiles of the error function are produced in the case of noise-free data. (B) Normalised log-likelihood surface for the logistic model, showing the maximum likelihood estimate (black dot) and both eigenvectors of the Fisher information matrix. $v_1$ corresponds to the eigenvector with the smallest eigenvalue. Note that axes are scaled relative to the maximum likelihood estimate $(\hat{\lambda}, \hat{R}_{\max})$.

profiled point, and the initially specified guess; in the case of a disparity between these three optimisations, the result with the largest log-likelihood is taken as the maximum.

We establish the identifiability of parameters in the logistic, radial-death, and Greenspan models using the profile likelihood method in Fig 5A, 5C and 5D using synthetic data generated from the Greenspan model (shown previously in Fig 1B). Both the logistic and radial-

death models are identifiable; parameter estimates can be established to within a two-sided 95% confidence interval. We also see that confidence intervals for the per-volume growth rate, λ, are consistent between each model. Results in Fig 5D show that parameters in the Greenspan model; specifically, $Q$ and $\gamma$, are non-identifiable, or one-sided identifiable. Estimates for the spheroid radius at which necrosis first occurs, $R_d$, are constrained to a two-sided 95% confidence interval, however the upper bound of this confidence interval corresponds to the maximum spheroid size observed, suggesting that $R_d$ is also only one-sided identifiable.

To explore whether parameters in the Greenspan model are identifiable in the limit of noise-free data, we produce profiles of the error function (Eq 21) in the hypothetical case that noise free observations are made so that $\sigma = 0$ μm. This is done in a similar manner to profiles of the log-likelihood function, albeit where the maximisation is replaced by a minimisation. Results are shown in Fig 5E. Our aim is to determine whether the parameters uniquely map to the data in the noise-free limit or, equivalently, whether there exist other parameter combinations in the vicinity of $\mathbf{p}_1$ where the error function is zero. Results in Fig 5E show that the error function has a clearly defined minimum, indicating that the parameters in the Greenspan model are theoretically identifiable in the noise-free limit.

**Fisher information.** For models with additive normal noise, the Fisher information matrix (FIM) for model $i$ is given by

$$F_i(\mathbf{p}) = [J_i(\mathbf{p})]^\top J_i(\mathbf{p}), \tag{23}$$

where $J_i$ is the Jacobian of model $i$, sometimes referred to as the *parameter sensitivity matrix* [49]. For spheroid radius data collected at $n$ time points, $J_i$ is an $n \times k_i$ matrix with elements

$$J_i(\mathbf{p}) = [\nabla_\mathbf{p} R_i(t^{(1)}; \mathbf{p}) \nabla_\mathbf{p} R_i(t^{(2)}; \mathbf{p}) \cdots \nabla_\mathbf{p} R_i(t^{(n)}; \mathbf{p})], \tag{24}$$

where $k_i$ is the number of parameters. The FIM is a $k_i \times k_i$ matrix related to the expected Hessian (and, therefore, the curvature) of the log-likelihood function (Eq 20) and least-squares cost function (Eq 21).

The rank of the FIM at the MLE relates to the number of identifiable parameter combinations [49]. We find that the FIM of models studied in Fig 5 (the logistic, radial-death, and Greenspan models) have full rank. This is consistent with results from the profiled error function (Fig 5E) where we found that, although model parameters were not practically identifiable from the available data, they are identifiable from noise-free data.

For many models the FIM may be of full rank but have a large condition number and is, therefore, close to singular. For example, we find that the condition number of the FIM for the Greenspan model is $\mathcal{O}(10^9)$, suggesting that the FIM is full rank and non-singular, but has at least one uninformative direction. We can also see this from profile likelihood results in Fig 5D, which show that $Q$ and $\gamma$ cannot be constrained to be within 95% confidence, indicating a large parameter variance and correspondingly close-to-singular FIM. Analysis of *model sloppiness* provides finer-grained information about identifiability by gaining insight from the full spectrum of the FIM (Fig 1D). In summary, such analysis establishes directions in parameter space that are *stiff*, i.e., identifiable from data; and those that are *sloppy*, i.e., non-identifiable.

We demonstrate the relationship between the log-likelihood surface and the eigenvectors of the FIM for the logistic model in Fig 5B. The direction defined by the eigenvector with the largest eigenvalue, $v_2$, points in the direction of steepest descent from the MLE. The direction defined by $v_1$, the eigenvector with the smallest eigenvalue, points in the direction of shallowest descent from the MLE. Should an eigenvalue tend to zero, the likelihood becomes flat in the direction of the corresponding eigenvector and parameters that lie on this contour are indistinguishable: this direction is *sloppy*.

In Fig 1D we show the log eigenvalues of the FIM for each model, scaled by the largest eigenvalue for each model. All models, aside from the Greenspan model and Ward and King's model have eigenvalues constrained over a relatively small number of decades. Greenspan's model has one eigenvalue much smaller in magnitude than the remaining, suggesting a single sloppy or uninformative direction. Similarly, Ward and King's model has two eigenvalues much smaller in magnitude than the remaining, suggesting two sloppy directions.

## Geometric analysis

Identifiability and model sloppiness analysis indicates that several parameters in the Greenspan model are not identifiable from spheroid radius measurements, however we cannot gain further information relating to the impact each non-identifiable parameter has on the features of the data. To study this further, we examine the geometric relationship between the Greenspan model, and the simplistic, identifiable, logistic and radial-death models.

We define a map from the parameters in model $i$, denoted $\mathbf{p}_i$, to parameters in the identifiable model $j$, denoted $\mathbf{p}_j$, in the least-squares sense such that

$$\mathbf{p}_j = \mathbf{f}_{ij}(\mathbf{p}_i), \tag{25}$$

where

$$\mathbf{f}_{ij}(\mathbf{p}_i) = \arg \min_{\mathbf{p}_j} \|\mathbf{m}_i(\mathbf{p}_i) - \mathbf{m}_j(\mathbf{p}_j)\|. \tag{26}$$

An interpretation of $\mathbf{p}_j = \mathbf{f}_{ij}(\mathbf{p}_i)$ is the maximum likelihood or least-squares estimate for the parameters in model $j$ if noise-free data from model $i$ is observed. To quantify goodness-of-fit, which we interpret as a measure of the correspondence between models, we compute the $R^2$ statistic

$$R^2 = 1 - \frac{\|\mathbf{m}_i(\mathbf{p}_i) - \mathbf{m}_j(\mathbf{f}_{ij}(\mathbf{p}_i))\|^2}{\|\mathbf{m}_i(\mathbf{p}_i) - E[\mathbf{m}_i(\mathbf{p}_i)]\|^2}, \tag{27}$$

where $E[\cdot]$ denotes the sample mean. In this work, we take the supremum of the log-likelihood function numerically using the Nelder-Mead algorithm over a region that covers the true parameters over several orders of magnitude [51].

In Fig 4B, we show noise-free data generated from the Greenspan model with $\mathbf{p}_1 = [Q, R_d, \gamma, \lambda]^\mathsf{T} = [0.8, 150, 1, 1]^\mathsf{T}$. In this case, we have good correspondence between the logistic and Greenspan models ($R^2 = 0.998$) with $\mathbf{p}_2 = [\lambda, R_{max}]^\mathsf{T} = [1.21, 316]^\mathsf{T}$. Despite both models sharing a parameter, $\lambda$, with an equivalent biological interpretation (the per volume growth rate), estimates for this parameter differ between models. This result highlights that parameter estimates are not necessarily directly comparable between models despite sharing a similar biological interpretation [21]. In our case, both the Greenspan and logistic models assume that cells proliferate exponentially at rate $\lambda$ for infinitesimally small spheroids, however the crowding functions differ between the models such that the logistic model does not capture the initial exponential growth phase seen in the Greenspan model (Fig 4B). Given that the bounded Gompertz and Greenspan models have comparable crowding functions, we see better agreement between estimates for $\lambda$ between these models ($R^2 = 0.99997$ with $\mathbf{p}_3 = [\lambda, R_{max}]^\mathsf{T} = [1.00, 317]^\mathsf{T}$). In all cases, estimates for the maximum radius, $R_{max}$, agree with the long-term solution of the Greenspan model calculated numerically ($R_{max} = 320$ μm).

To study between-model sensitivities, we compute the Jacobian matrix of $\mathbf{f}_{ij}(\mathbf{p}_i)$, denoted

$$J_{ij} = \frac{\partial \mathbf{f}_{ij}}{\partial \mathbf{p}_i}. \tag{28}$$

We compute $J_{ij}(\mathbf{p}_i)$ numerically, using a finite difference approximation that is implemented in a standardised algorithm that is robust to numerical noise introduced from the optimisation algorithm used to calculate $\mathbf{f}_{ij}(\mathbf{p}_i)$ [52]. The rows of $J_{ij}(\mathbf{p}_i)$ correspond to the gradients of each element in $\mathbf{p}_j$, denoted by $\nabla \mathbf{p}_j^{(k)}$. These vectors are normal to, and hence define, a hyperplane in model $i$ parameter-space that locally give identical estimates of $\mathbf{p}_j^{(k)}$ in the vicinity of $\mathbf{p}_i$. For example, we can use $J_{ij}(\mathbf{p}_i)$ to visualise the parameter combinations in the Greenspan model that give identical estimates of $\lambda$ and $R_{\max}$ in the logistic and bounded Gompertz models.

While geometrically useful, it is difficult to interpret the elements of $J_{ij}(\mathbf{p}_i)$ directly as the scales of each parameters differ significantly, even within each model. Therefore, we introduce the sensitivity matrix of $\mathbf{p}_j = \mathbf{f}_{ij}(\mathbf{p}_i)$, denoted

$$S_{ij}(\mathbf{p}_i) = \mathrm{diag}(\mathbf{p}_i)^{-1} J_{ij}(\mathbf{p}_i) \, \mathrm{diag}(\mathbf{p}_j). \tag{29}$$

The $(k_1, k_2)$ element of the sensitivity matrix is given by

$$S_{ij}^{(k_1,k_2)}(\mathbf{p}_i) = \frac{\mathbf{p}_j^{(k_1)}}{\mathbf{p}_i^{(k_2)}} J_{ij}^{(k_1,k_2)}(\mathbf{p}_i), \tag{30}$$

and can be interpreted as the relative increase in parameters in model $j$ due to increases in parameters in model $i$. For the map from the Greenspan to logistic model we have

$$S_{12}(\mathbf{p}_1) = \begin{matrix} & Q & R_{\mathrm{d}} & \gamma & \lambda & \\ \begin{bmatrix} -0.0156 & -0.0493 & 0.105 & 0.989 \\ 0.901 & 1.01 & -0.304 & 0.0141 \end{bmatrix} & \begin{matrix} \lambda \\ R_{\max} \end{matrix} \end{matrix}. \tag{31}$$

Here, we see a near one-to-one correspondence in $\lambda$ between models and, for example, see that a 1% increase in $Q$ in the Greenspan model is associated with a 0.0156% decrease in $\lambda$ in the logistic model. Furthermore, this analysis demonstrates that, roughly speaking, the parameter subset $(Q, R_{\mathrm{d}}, \gamma)$ corresponds primarily to the maximum spheroid size. We draw this conclusion based on the relative sizes of elements in each row of $S_{12}(\mathbf{p}_1)$ (Eq 31). In other words, there exists a trivariate function of $(Q, R_{\mathrm{d}}, \gamma)$ that maps to $R_{\max}$, and by extension the likelihood. This explains why the $(Q, R_{\mathrm{d}}, \gamma)$ are practically non-identifiable Fig 5, and since $R_{\max}$ is identifiable we expect that the relationship between $(Q, R_{\mathrm{d}}, \gamma)$ is also identifiable while the individual parameters are not. This analysis is similar to existing techniques used to establish identifiable parameter combinations using the likelihood, through profiling [53, 54] or the FIM [55]. From Eq (31) we also see that the per-volume proliferation rates correspond in each model. This latter observation is entirely consistent with profile likelihood analysis in Fig 5, where we see that estimates for $\lambda$ in the Greenspan model are identifiable.

**Geometric analysis using the logistic model.** As the Greenspan model has only four parameters, one of which is practically identifiable (Fig 5D), we can visually explore the geometric link between the Greenspan and logistic models to provide insight into the relationship between these models. To do this, we fix the identifiable parameter at the true value, $\lambda = 1$ h$^{-1}$, and numerically compute the coordinates of the remaining parameter values $[\gamma, Q, R_{\mathrm{d}}]^{\mathsf{T}}$ for which $\mathbf{f}_{12}(\mathbf{p})$ is constant; that is, parameters in the Greenspan model that map to the same set of parameters in the logistic model as at the true value $\mathbf{p}_1 = [Q, R_{\mathrm{d}}, \gamma, \lambda]^{\mathsf{T}} = [0.8, 150, 1, 1]^{\mathsf{T}}$. We

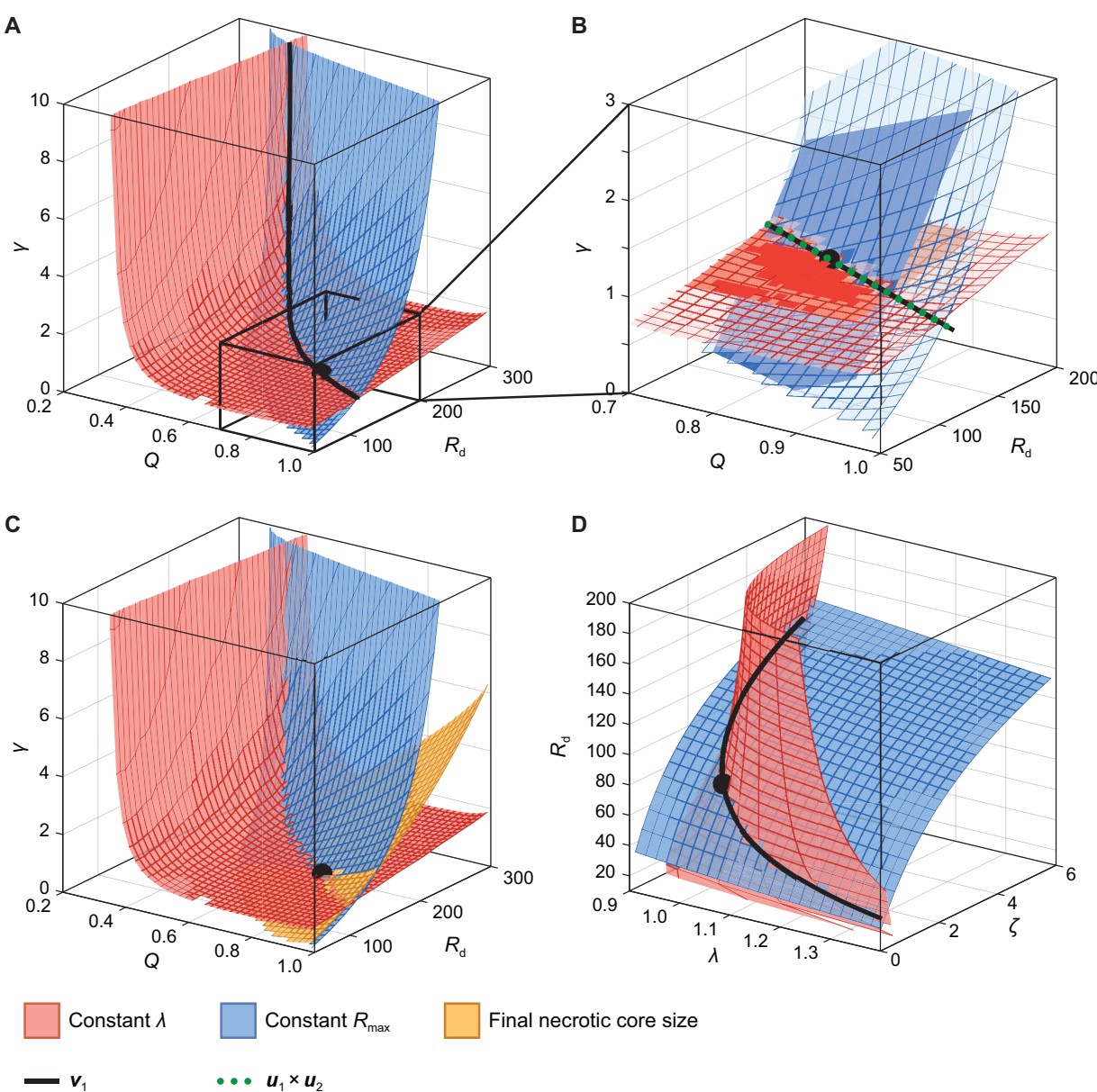

**Fig 6. Visualisation of the between-model geometry between the Greenspan, logistic, and radial-death models.** Surfaces correspond to manifolds in the parameter space of the (A–C) Greenspan and (D) radial-death models that give rise to (red) constant volumetric growth rate, λ, and (blue) constant maximum spheroid size, $R_{max}$, when compared to the logistic model. In all cases, the parameter set used for the comparison is shown as a black disc. In (A,D), black curves correspond to the solution to Eq (32), and give the curve of near-constant likelihood. Note that this curve is visualised in front of the manifolds for clarity. In (B), we compare the manifolds (semi-transparent) to a linearisation corresponding to two planes normal to the rows of the model-map Jacobian (Eq (33)). We show the sloppy direction $\mathbf{v}_1$ (black) and the cross product of the model-map Jacobian rows (green dotted). In (C) we show a third manifold corresponding to constant final necrotic core size, which is sufficient to identify the parameters in the Greenspan model.

show the associated two-dimensional manifolds that give a constant proliferation rate, λ, and constant maximum spheroid size, $R_{max}$, in Fig 6A. The intersection of these manifolds corresponds to the one-dimensional manifold that give a set of parameters in the Greenspan model that map to the same solution curve to the logistic model in the least-squares sense. As the logistic model has good correspondence to the Greenspan model ($R^2 = 0.998$), we expect that

this intersection corresponds to a curve of near-constant likelihood; parameter sets that lie on this line are indistinguishable, leading to non-identifiability of individual parameters.

The dimensionality of this manifold corresponds to the number of uninformative or sloppy directions in the Greenspan model, observed in Fig 1D. However, as the corresponding eigenvalue was small but non-zero, this curve is not a curve of *constant* likelihood, but rather a curve of *near-constant* likelihood. We consider that the eigenvector associated with the smallest eigenvalue of $F_1(\mathbf{p})$ (the FIM of the Greenspan model), denoted at each point by $\mathbf{v}_1(\mathbf{p})$, defines a sloppy direction in parameter space; that is, a direction in which the model is relatively insensitive. Starting at the true values, we follow the sloppy direction through parameter space by solving the ODE

$$\frac{d\mathbf{p}}{dt} = \mathbf{v}_1(\mathbf{p}),\qquad(32)$$

subject to $\mathbf{p}(0) = \mathbf{p}_1$, as shown in Fig 6A. As expected, this curve follows the intersection of the constant $\lambda$ and constant $R_{max}$ manifolds. In Fig 6B we demonstrate that the sloppy direction is orthogonal to both manifolds using the linearisation of each manifold formulated from the model-map Jacobian,

$$J_{12}(\mathbf{p}_1) = \begin{bmatrix} \nabla\lambda(\mathbf{p}_1) \\ \nabla R_{max}(\mathbf{p}_1) \end{bmatrix} = \begin{bmatrix} \mathbf{u}_1 \\ \mathbf{u}_2 \end{bmatrix}.\qquad(33)$$

Here, $\mathbf{u}_1$ gives the gradient of $\lambda$ in the logistic model with respect to the parameters in the Greenspan model, and similar for $\mathbf{u}_2$ to $R_{max}$. Therefore, $\mathbf{u}_1$ and $\mathbf{u}_2$ are normal to the constant $\lambda$ and constant $R_{max}$ manifolds at $\mathbf{p}_1$, and, therefore, define a tangent plane to each manifold at $\mathbf{p}_1$. In Fig 6B we show that the intersection of both tangent planes, given by the vector cross product $\mathbf{u}_1 \times \mathbf{u}_2$, corresponds to the sloppy direction $\mathbf{v}_1$.

We have established that data from outer radius measurements are insufficient to identify the parameters in the Greenspan model. Rather we can only constrain parameter estimates to a one-dimensional line that corresponds to the intersection of two two-dimensional manifolds. These results are consistent with our previous work [14, 18], where we demonstrate that measurements of the inner structure of spheroids, specifically, the necrotic core size, are required to identify parameters. We explore this in our geometric framework by considering a third two-dimensional manifold in the parameter space corresponding to realisations of the Greenspan model that give identical measurements of the necrotic core size at the conclusion of our synthetic experiment ($t = 21$ d). In Fig 6C we show that, as expected, the intersection of three two-dimensional manifolds corresponds to a zero-dimensional point, indicating parameter identifiability.

**Geometric analysis using the radial-death model.** The radial-death model, having three parameters, sits between the logistic and Greenspan model with respect to model complexity and identifiability. Therefore, we can apply the radial-death model to aid interpretation of parameters in the Greenspan model, and also learn about features of the radial-death model using the logistic model.

The sensitivity matrix for the map from the Greenspan to the radial-death model is given by

$$S_{15}(\mathbf{p}_1) = \begin{array}{c} \begin{array}{cccc} Q & R_d & \gamma & \lambda \end{array} \\ \begin{bmatrix} -0.005 & -0.00501 & -0.0107 & 1.07 \\ 0.348 & 0.0848 & 1.01 & 0.835 \\ 1.08 & 1.04 & 0.222 & -0.0727 \end{bmatrix} \begin{array}{c} \lambda \\ \zeta \\ R_d \end{array} \end{array}.\qquad(34)$$

First, we see a near one-to-one correspondence between the per-volume growth rate, $\lambda$, in each model. We expect this, as both models include a finite period of time where spheroid growth is exponential. Secondly, we see that $\gamma$ and $\lambda$ in the Greenspan model have a near one-to-one correspondence to $\zeta$ in the radial-death model. Finally, we see a near one-to-one correspondence between $Q$ and $R_d$ in Greenspan's model and $R_d$ in the radial-death model. Although $R_d$ has a similar interpretation in both models, in the radial-death model it must capture both the second phase of growth inhibition and the third phase of necrosis, both of which relate to $Q$ and $R_d$.

We study the map between the radial-death and Greenspan models at $\mathbf{p}_5 = \mathbf{f}_{15}(\mathbf{p}_1)$, the parameters in the radial-death model we find to be equivalent to those in the Greenspan model (Fig 4). The sensitivity matrix is given by

$$S_{52}(\mathbf{p}_5) = \begin{array}{c} \begin{array}{ccc} \lambda & \zeta & R_d \end{array} \\ \begin{bmatrix} 0.874 & 0.126 & -0.0506 \\ 0.519 & -0.52 & 1.01 \end{bmatrix} \begin{array}{c} \lambda \\ R_{\max} \end{array} \end{array}. \tag{35}$$

The maximum spheroid size, $R_{\max}$ is sensitive to all parameters in the radial-death model, whereas the per-volume growth rate, $\lambda$, has a near one-to-one correspondence between models. While the radial-death model is identifiable, we still see that the sloppiest direction corresponds to the intersection of the constant $\lambda$ and constant $R_{\max}$ manifolds defined by the map from the radial-death to logistic models (Fig 6D). In this case, we solve Eq (32) using the FIM for the radial-death model, and show the resultant curve within a 95% likelihood-based confidence region in the three-dimensional parameter space.

**Predicting non-identifiability.** Whereas traditional identifiability and sloppiness analysis provides directions (if any) in the parameter space to which the model output is insensitive, our geometric analysis provides information about the sensitivity of model *features* to changes in parameters. This allows us to predict non-identifiability using known identifiability results for the simpler, phenomenological models. For instance, for $R(t)/R_{\max} \ll 1$ the solution of the logistic model corresponds to exponential growth

$$R(t) \sim R_0 \exp\left(\frac{\lambda t}{3}\right), \tag{36}$$

which does not depend on $R_{\max}$. We conclude that $R_{\max}$ is non-identifiable from early-time data.

In Fig 7, we establish the practical identifiability of the logistic, radial-death, and Greenspan models in the case that 22 equally-spaced early-time observations are made for $0 \le t \le 5$ d, using profile likelihood analysis. Again, we generate synthetic data from the Greenspan model, however reduce the variance of observations $\sigma = 2$ μm so that the confidence interval for estimates of $\lambda$ in the logistic model are comparable to those in Fig 5, where measurements are taken for $0 \le t \le 21$ d. As expected, $R_{\max}$ is non-identifiable (specifically, $R_{\max}$ is one-sided identifiable, as we can establish that the maximum spheroid size must be greater than the observed size of spheroids).

Geometric analysis of the map between the radial-death and logistic models (Eq (35)) indicates that $\lambda$ in the logistic model is insensitive to changes in $(\zeta, R_d)$ in the radial-death model. Therefore, since only $\lambda$ is identifiable from early-time data, we expect that $(\zeta;, R_d)$ are now non-identifiable. We see this in profile likelihood analysis for the radial-death model in Fig 7C. From the geometric analysis we were able to determine that information about $(\zeta, R_d)$ is contained in late-time data. Similarly, geometric analysis of the map between the Greenspan and

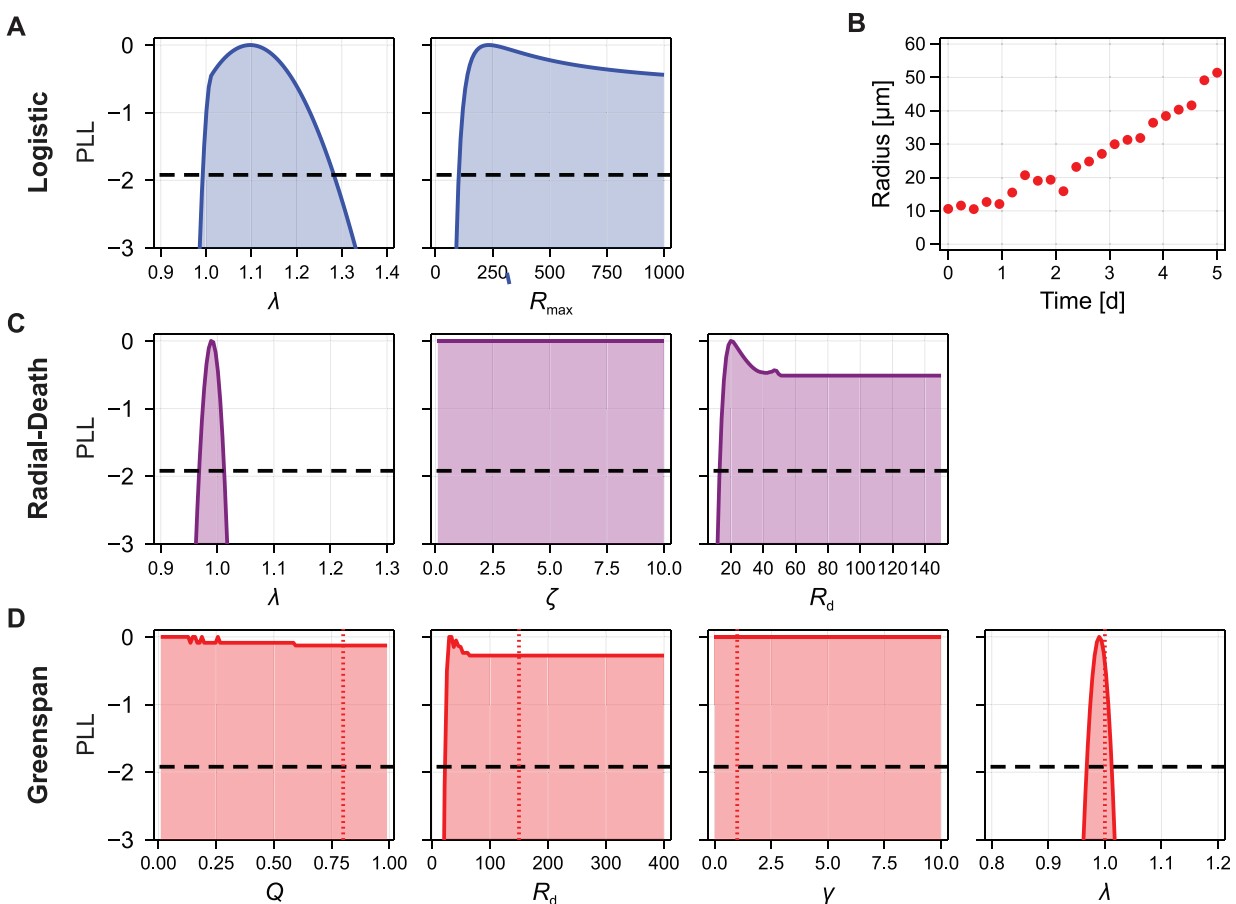

**Fig 7. Identifiability analysis for the logistic, radial-death, and Greenspan models using early-time data.** (A,C,D) Profile likelihood for the parameters in each model, using synthetic data generated from the Greenspan model with standard deviation σ = 2 μm. Synthetic data is shown in (B). Also shown is the threshold for an approximate 95% confidence interval (black-dashed). For the Greenspan model, we additionally show the parameter values used to generate the synthetic data.

logistic models (Eq (31)) established that changes in $(Q, R_d, \gamma)$ correspond to changes in $R_{max}$, but not $\lambda$. Again, our analysis indicates that information about $(Q, R_d, \gamma)$ would not be available from early-time data. These results are consistent with profile likelihood analysis (Fig 7D), which shows that profiles for $(Q, R_d, \gamma)$ are flat; while $(Q, R_d, \gamma)$ are also non-identifiable from observations up to $t = 21$ d, there is less information about the parameters from early-time data.

**Gaining insights from complex, non-identifiable models.** For interpreting spheroid radius data, the Ward and King model performs poorly compared to the other models considered based on AIC and model sloppiness analysis (Fig 1C and 1D). As significantly simpler models can provide comparable fits, we conclude that the information contained in outer radius measurements is insufficient to identify the eight parameters in the (already simplified) Ward and King model. The reason for number of parameters in the Ward and King model is the number of biological mechanisms it includes: it is the only model we consider that explicitly incorporates the consumption of nutrients by cells, the passage of nutrients from the spheroid exterior inside the spheroid, cell death, and the initial spatial distribution of cells inside the spheroid.

We study the sensitivity of parameters in the Ward and King model to features indicated by the logistic model at $\mathbf{p}_6 = \mathbf{f}_{16}(\mathbf{p}_1)$ where an initial guess of $\bar{\mathbf{p}}_6 = [10, 1.5, 1.4, 0.4, 0.35, 0.07, 6 \times 10^5, 3 \times 10^4, 0.1]^\top$ is used in the optimisation routine. The

sensitivity matrix is given by

$$
S_{62}(\mathbf{p}_6) = \begin{array}{cccccccc}
\lambda & \delta & c_1 & c_2 & \alpha & D_p & Q_p & p_0 \\
\end{array}
$$

$$
S_{62}(\mathbf{p}_6) = \begin{bmatrix}
0.641 & 0.135 & -0.0318 & 0.0474 & 0.0247 & -0.0143 & 0.0129 & -0.0452 \\
0.563 & -0.5 & 0.234 & -0.368 & -0.515 & -0.0158 & 0.00482 & 0.1
\end{bmatrix}
\begin{array}{l} \lambda \\ R_{\max} \end{array} \quad (37)
$$

Table 1 contains a summary describing the biological interpretation and biological mechanism associated with each parameter. First, we see a correspondence in the per-volume growth rate between models. As expected, increasing the cell per-volume growth and death rates have opposing effects on the maximum spheroid size. Interestingly, we find that increasing the relative nutrient consumption rate, $\alpha$, has little impact on the the growth rate, but does cause a decrease in the maximum spheroid size. Further, we see that increasing the nutrient threshold for growth inhibition, $c_1$, causes an *increase* in the maximum spheroid size; if cells require more oxygen to proliferate, they do so more slowly but this may, overall, yield larger spheroids.

Given that there are eight parameters in the Ward and King model, it is difficult to visualise the geometry of the map from the Ward and King model to the logistic model. However, we can apply the Jacobian of the Ward and King to logistic model-map, $J_{62}(\mathbf{p}_6)$, to interpret the model sloppiness results for the Ward and King model in Fig 1D. We denote the rows of $J_{62}$ as $\mathbf{u}_1$ and $\mathbf{u}_2$, corresponding to gradients with respect to $\lambda$ and $R_{\max}$, respectively. Recalling that the eigenvalues of the FIM correspond to the curvature of the expected log-likelihood in the direction of each corresponding eigenvector, we interpret the relative magnitude of each eigenvalue as a measure of how informative the corresponding direction is. In Table 2 we tabulate the eigenvalues of the FIM and the dot products between the corresponding eigenvectors and unit vectors in the direction of $\mathbf{u}_1$ and $\mathbf{u}_2$. Dot products close in absolute value to zero indicate orthogonality, dot products close in absolute value to one indicate a correspondence between the directions. These results confirm that sloppy or uninformative directions are orthogonal to directions that correspond to large changes in $\lambda$ and $R_{\max}$.

The elements of the Jacobian of the model-map relate to *absolute* changes in the parameters, whereas elements of the sensitivity matrix relate to *relative* changes in the parameters. Therefore, we can use the rows of the sensitivity matrix, denoted $\mathbf{s}_1$ and $\mathbf{s}_2$ to move around the parameter space of the Ward and King model to achieve relative changes in the volumetric growth rate, $\lambda$, and relative changes in the maximum spheroid radius, $R_{\max}$, respectively (full details are available in S1 File). We demonstrate this in Fig 8A and 8B, where we choose

**Table 2. Orthogonality between the model-map and uninformative directions.** We tabulate the eigenvalues of the FIM relative to the largest eigenvalue, and the dot product between each corresponding eigenvector and rows of the Jacobian of the Ward and King to logistic model-map.

| Eigenvalue | $\mathbf{v} \cdot \hat{\mathbf{u}}_1$ | $\mathbf{v} \cdot \hat{\mathbf{u}}_2$ |
|---|---|---|
| $2.18 \times 10^{-18}$ | $3.06 \times 10^{-8}$ | $6.27 \times 10^{-9}$ |
| $2.41 \times 10^{-15}$ | $-1.13 \times 10^{-7}$ | $1.99 \times 10^{-7}$ |
| $4.04 \times 10^{-6}$ | $-6.38 \times 10^{-2}$ | $3.88 \times 10^{-3}$ |
| $6.74 \times 10^{-5}$ | $-4.18 \times 10^{-1}$ | $4.09 \times 10^{-2}$ |
| $1.51 \times 10^{-4}$ | $3.26 \times 10^{-2}$ | $1.24 \times 10^{-2}$ |
| $3.73 \times 10^{-4}$ | $-7.03 \times 10^{-1}$ | $3.31 \times 10^{-2}$ |
| $1.12 \times 10^{-2}$ | $2.05 \times 10^{-1}$ | $-2.21 \times 10^{-1}$ |
| $1.0$ | $-5.33 \times 10^{-1}$ | $9.74 \times 10^{-1}$ |

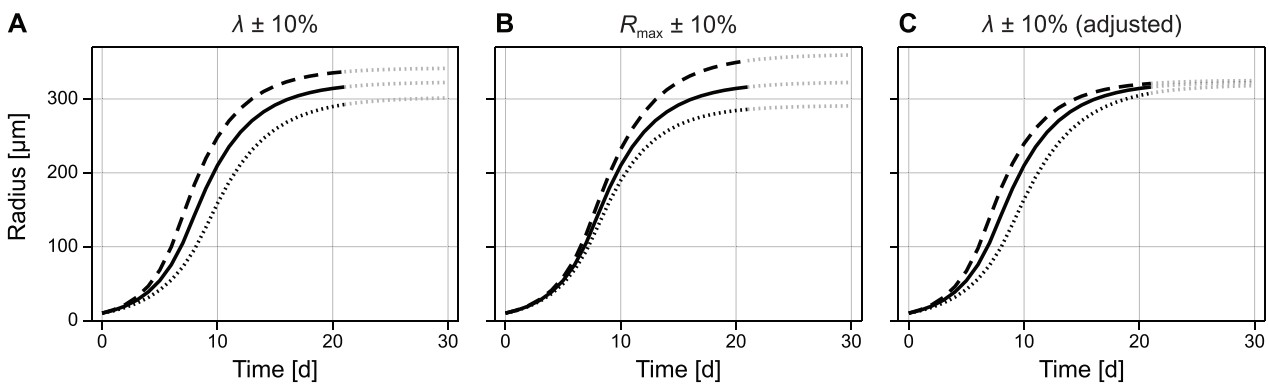

**Fig 8. Using the sensitivity matrix of the model-map to achieve relative changes in model features.** We apply the rows in the sensitivity matrix to adjust the parameters in the Ward and King model to achieve approximate relative changes in the parameters in the logistic model. In (A–B) we move in the direction of each row of the corresponding row of the sensitivity matrix; in (C) we move in direction of a projection of row 1 orthogonal to row 2, resulting in relative changes to the volumetric growth rate, but not to maximum spheroid radius.

adjusted parameters in the Ward and King model to achieve a 10% relative increase and decrease in both $\lambda$ and $R_{max}$.

Since $\mathbf{s}_1$ and $\mathbf{s}_2$ are not orthogonal, moving in the relative direction of $\mathbf{s}_1$ results in an increase to *both* $\lambda$ and $R_{max}$. However, and potentially more usefully, we can move in the direction of an orthogonal projection of $\mathbf{s}_1$ onto $\mathbf{s}_2$ to achieve, approximately, a relative change in $\lambda$ without changing $R_{max}$. We demonstrate this in Fig 8C, showing an increase in the volumetric growth rate but a much smaller change in the maximum spheroid radius compared to results in Fig 8A.

## Discussion

The nexus between model complexity and data quantity and quality is an ongoing challenge in computational biology that is often resolved subjectively rather than objectively. While new experimental technologies are rapidly increasing the detail and resolution obtainable in biological data, mathematical models can always be made arbitrarily complex. On the other hand, data is often limited in light of the biological questions that are posed. Identifiability and sloppiness analyses have been developed to harmonise model and data complexity, to guide model selection and reduction, in order to ensure parameter identifiability [19, 56]. However, the complex, highly-detailed, heavily-parameterised models that are commonplace in mathematical and computational biology are often required to answer important biological questions: a model of tumour spheroid growth must incorporate nutrient dependencies to provide insight into the role nutrients play on growth [8, 57]. As we show, for some data only simple phenomenological models, such as the logistic and Gompertz growth models, are those that are identifiable. These models can provide excellent agreement to experimental data, allow the comparison and interpretation of experiments, however not being constructivist, provide only limited insights into underlying biological mechanisms.

In many cases, simple phenomenological models produce a goodness-of-fit on par with that of a complex mechanistic model (Fig 1B and 1C). As a result, traditional model selection methodology will favour the simplicity and identifiability of the simple model, penalising the number of parameters in the complex model. Where the non-identifiable parameters in complex mechanistic models carry direct biological interpretations (the nutrient sensitivity, for instance) of prime interest to experimental scientists and biologists, the identifiable parameters in the simple model carry interpretations relating to *features* of the data (the early-time rate of change or the maximum spheroid size, for instance). In this work, we utilise this key difference

to draw biological insight from complex mechanistic models by studying the geometry of a map from the parameters in the complex model to those in the identifiable surrogate. One interpretation of our approach is to provide an intermediate mode of interpretation that sits between the model parameters and the likelihood (or other goodness-of-fit metric) that is traditionally studied in identifiability and model sloppiness analysis. In contrast to studying the sensitivity of the model in terms of the *overall* fit, we effectively decompose the fit into *features* and study the sensitivity of model parameters to these features. This approach enables us leverage mechanistic modelling to gain insights from data that would otherwise be lost if a one-dimensional goodness-of-fit metric, such as the likelihood, were to be studied directly.

We demonstrate our approach by analysing common models and typical data of tumour spheroid growth. Mathematical models of tumour spheroid experiments range from the simplistic—however routinely and effectively applied—logistic growth models [11, 58], to spatial models that can capture the density of arbitrary numbers of cell and nutrient species [13, 17, 59, 60], and to individual-based models that describe the individual behaviour of every cell in the spheroid [61–63]. Despite the complexity of even this simple experimental model of tumour growth, data often comprise only measurements of overall tumour spheroid size. More complicated experimental systems, such as *in vivo* vascularised tumour growth, are accompanied by a corresponding menagerie of complex models [64–66], however data from these experiments can be even more limited, noisy, or sparse in comparison with experimental models of avascular tumour growth [18]. Even the relatively simple Greenspan model, which comprises only four unknown parameters, is non-identifiable without measurements of inner spheroid structure [14]. Our goal in this work is to draw insights from such models with complexity mismatched to that of the available data.

The model-data relationship is typically explored with structural or practical identifiability analysis [26]; the former in an infinite-data, model-only frame of reference, the latter in consideration of the noisy observation process that ties the model to the data. While we first establish the practical identifiability of each model, our geometric analysis does not fall into either of these classifications for a number of reasons. First, the model-map is defined in the least squares sense, and does not explicitly incorporate data. Secondly, as the surrogate model is not necessarily nested within the complex model of interest, the two are not equivalent in a meaningful infinite-noise-free-data limit. As a consequence, if the complex model is considered reality, the surrogate model produces predictions that are biased. In the context of data, we see this as an advantage as even complex models are by definition abstractions of reality. As we utilise the surrogate model to characterise features of the data, our approach is overall robust to this bias. We demonstrate this by using the logistic model as a surrogate for the Greenspan model in the main text, despite the bounded Gompertz model having a crowding function far more similar to that of the Greenspan model. Analysis using the Gompertz and Richards models (S1 File) is similar to that using the logistic model.

A limitation of FIM-based identifiability and sloppiness analysis, and our sensitivity-matrix-based geometric analysis, is a restriction to providing only local information. Effectively, these techniques relate to a quadratic approximation and linearisation, respectively, about the MLE (or parameter values otherwise under consideration) of the complex model and are consequentially sensitive in cases where the corresponding likelihood is multimodal. While the manifolds relating to the map between the logistic and Greenspan models (Fig 6A and 6B) are locally linear near the parameter combination of interest, globally the manifold relating to λ appears hyperbolic. Different points on the constant-likelihood curve have the potential to produce substantially different sensitivity matrices. One approach to address this is to incorporate prior knowledge to regularise the parameter fitting problem. Recent work considers identifiability and sloppiness analysis based directly on the parameter covariance

matrix estimated from Bayesian methods such as Markov-chain Monte-Carlo to provide an overall snapshot of the global parameter sensitivities [35]. However, we expect this approach to be problematic in our geometric framework, since the model-map is based on an equivalence between models that may only apply locally in the vicinity of the points used to compute the between-model sensitivity matrix.

The relatively small number of unknown parameters in the Greenspan model allow us to visually explore the geometry of the parameter space using surrogate models, providing insight into non-linearities that are not captured by the model-map sensitivity matrix. However, in contrast to traditional identifiability analysis where parameters are generally classified as identifiable or not, the model-map sensitivity matrix has the ability to further classify non-identifiable parameters by which feature they relate to. In the vicinity of the parameter values of interest, this classification can allow for graphical geometric analysis even for models with more than three unknown parameters, by decomposing the parameter space into low-dimensional subsets that relate to individual features. For example, in the Ward and King model, the three parameters with the strongest correspondence to the maximum spheroid size, $(\lambda, \alpha, \delta)$, can be prioritised for further analysis ahead of the full, eight-dimensional parameter space [29].

Aside from providing insight into the sensitivity of model features to parameters and predicting non-identifiability, we provide a simple demonstration of how the model-map relationship can be used to move in the parameter space to produce changes to specific model features (Fig 8C). Akin to moving in the direction of the sloppiest direction, these results show how to constrain movements in the parameter space to model feature manifolds. For heavily-parameterised models that are difficult to calibrate (perhaps, for example, due to multi-modal likelihoods), constraining movements in the optimisation algorithm used for model calibration to these manifolds allows successive matching of model features: for instance, first moving to parameter combinations that produce the desired maximum spheroid radius, and then moving on this manifold to match the growth curve shape and scale. More generally, applying surrogate models that are themselves candidate models raises interesting possibilities for future analysis. Generalisations of the logistic model, such as the three-parameter Richards model, provide a low-dimensional summary of model behaviour that can be characterised using machine learning or Gaussian processes [67]. Building up a global model-map between the complex and surrogate models is another approach to overcome the localisation limitation of our methods, and could be computationally advantageous in the case of computationally expensive complex models.

Experimental data are often limited in light of the biological questions posed of them. Likewise, in the mathematical and modelling literature, complex models are numerous and often more suitable to biological questions of interest, yet can be ill-suited for parameterisation from the available data. In this work, we develop a geometric analysis to gain insights from complex, non-identifiable models using simple surrogate models with parameters that relate to features in the data. We expect our analysis to apply to any hierarchy of non-identifiable and identifiable models of biological systems.

## Supporting information

**S1 File. Supporting material document.**
(PDF)

## Acknowledgments

We thank Nikolas Haass and Gency Gunasingh for training A.P.B. to perform the tumour spheroid experiments that motivated this work.

## Author Contributions

**Conceptualization:** Alexander P. Browning, Matthew J. Simpson.

**Formal analysis:** Alexander P. Browning, Matthew J. Simpson.

**Funding acquisition:** Matthew J. Simpson.

**Investigation:** Alexander P. Browning, Matthew J. Simpson.

**Methodology:** Alexander P. Browning, Matthew J. Simpson.

**Project administration:** Alexander P. Browning, Matthew J. Simpson.

**Software:** Alexander P. Browning.

**Supervision:** Matthew J. Simpson.

**Writing – original draft:** Alexander P. Browning.

**Writing – review & editing:** Alexander P. Browning, Matthew J. Simpson.

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
