## [Decision Letter · Decision Letter 0]

7 Nov 2022

Dear Professor Simpson,

Thank you very much for submitting your manuscript "Geometric analysis enables biological insight from complex non-identifiable models using simple surrogates" for consideration at PLOS Computational Biology. As with all papers reviewed by the journal, your manuscript was reviewed by members of the editorial board and by several independent reviewers. The reviewers appreciated the attention to an important topic. Based on the reviews, we are likely to accept this manuscript for publication, providing that you modify the manuscript according to the review recommendations.

Sincerely,

Nicholas Mancuso

Guest Editor

PLOS Computational Biology

Douglas Lauffenburger

Section Editor

PLOS Computational Biology

Reviewer's Responses to Questions

**Comments to the Authors:**

Reviewer #1: In their manuscript "Geometric analysis enables biological insight from complex non-identifiable models using simple surrogates", Browning and Simpson develop a new approach for investigating complex mechanistic models which are often non-identifiable by studying maps to simpler models with fewer parameters that are identifiable. I very much enjoyed reading this work - the approach seems to be technically reasonable, the manuscript is very well-written and the new method described here is highly original and looks to have quite a few promising applications.

I have some minor comments as well as a few general questions. These are mostly meant to be suggestions but it would be good if the authors could discuss these questions, even if only briefly.

1. Non-identifiability can indicate that some information about the system is irretrievably lost.

a. The approach is based on the implicit assumption that at least some aspects of the processes represented in the more complex models can be inferred by fitting the model to the data available. I don't think that this is necessary true, even if we ignore non-trivial examples of models that are structurally non-identifiable. Looking at the example of a growing tumour spheroid investigated in this study it is a legitimate question to which extent the complex underlying processes of a growing tumour spheroid can possibly contained in highly simplistic experimental data only consisting of the tumour spheroid radius over time. For the Ward & King model this comes down to the assumption that we can gain essential information regarding the concentration profile of oxygen within the spheroid, just by observing how quickly the tumour expands.

b. The authors might argue that the question how much can be learnt about the parameters of a complex model from simple data is exactly what their approach addresses. But it is very likely that the dependency of a complex process such as the development of an oxygen gradient on the data will respond very sensitively if the model is changed. So another implicit assumption is that the complex model is a better or more accurate representation of the system than alternative models.

2. a. I disagree with the authors' presentation of the logistic model (and other related models) as purely phenomenological. The logistic model has a clear mechanistic interpretation of individuals competing for space.

b. This matters because the authors state that the role of surrogate models like the logistic model is to identify "features" of the data. In the case of the spheroid data as far as I understand they refer to the growth rate and maximal radius. But the logistic model is more than a tool for inferring growth rate and maximal radius, it is also the result of a specific modelling decision of competition (in contrast to alternatives such as the whole class of Richards' models)

c. In my opinion, it would be better to present the models in this paper as a hierarchy of mechanistic models - logistic and related models (Gompertz, Richards') explain the observed growth of tumour spheroid as the result of competition without explicitly representing its underlying cause (such as competition for nutrients or oxygen).

3. The "complex" models in this papers are still quite simple and have relatively few parameters - the authors themselves mention some examples from the literature. It would be good if the authors could discuss if they expect that between-model jacobians and sensitivities would still useful for much more complicated models (of course, these models would most likely have the issue mentioned in 1.).

Minor comments:

p.3, above equation (1): "In practise" -> maybe "In practice"?

p.3, last paragraph: "We generate synthetic radius measurements... to the complex multiphase model of Ward and King [15,16]" - repeated in similar words on page 6.

p.5, "local-curvature" -> maybe "local curvature"?

p.5, "Geometrically, the FIM is related to the Hessian of the log- likelihood function and so contains information about the directions in parameter space in which the log-likelihood (and therefore the model) is both sensitive and insensitive [26]" - this sentence is unclear, what does "both sensitive and insensitive" mean? (although it becomes somewhat clearer after reading the next sentence)

p.6, Figure 2b: The figure shows a map from a non-identifiable model 3 to another non-identifiable model 2 which is then mapped to an identifiable model 1. As far as I understand in this manuscript the authors have only studied maps of one non-identifiable model to another identifiable model. Would it make sense to extend this process by constructing maps between non-identifiable models?

p.6, third line from the bottom: "Aside from the requirement that models are deterministic..." - Is this constraint necessary? I do actually think that this approach could be extended to stochastic models.

p.7: "The choice of R_0 = 10μm is made to ensure that the simple models we consider are identifiable." - It is not clear to me how identifiability depends on the initial condition.

p.7, last few sentences above "Model 1." - I think that the authors might consider simplifying this paragraph. I think it would be clearer to state that time and (if applicable space) are deliberately not non-dimensionalised.

p.9: "The logistic model and its generalisations (including the Gompertz and Richards models [42]) are purely phenomenological; they are not explicitly constructed from biological mechanisms by which overall growth is inhibited and eventually ceases." - as mentioned above in general comment 2. I don't think that this is true, it is possible to derive logistic growth as the result of competition for space.

p.9: "While logistic growth is commonly used to describe the time-evolution of spheroid volume [11], we find that logistic growth in the spheroid radius is more consistent with our synthetic data": I am not sure if I fully understand this sentence. Do the authors want to say that the logistic growth model is more commonly used for modelling tumour volume rather than radius?

p.11: "However, the logistic model differs in that it does not include a discrete initial phase where growth is exponential." - I am not sure if "discrete" is the right word here? I assume that this is in comparison with the bounded Gompertz model where the radius grows exponentially for 0 < R(t) < R_max / exp(1) ?

p.12, below eq. (12): Although equation (14) can, of course, be derived without explicitly stating the rate g of cells entering the necrotic core it would be good to state some characteristics, for example, that g(V_1)=0 if R(t)<r_d?

p.12, eq. (14): Also, in eq. (14) I assume that the terms involving R(t)-R_d vanish for R(t)<r_d? be="" if="" mentioned="" should="" so="" somewhere.="" this="">

p.13, below eq. (16): I find the description of the Ward & King model a bit terse. Without knowing the model the role of the "cellular material" is unclear, and, similarly, what it means that it is "assumed to have the same density as living cells". Also, for readers not familiar with moving boundaries, eq. (17) might be hard to understand without a bit more explanation.

p.14: under "Profile likelihood": "constrain" -> constraint

p.14, eq. (22): It would be helpful to remind the readers that \\hat{p}_i is the maximum-likelihood estimator when all parameters are varied.

p.15, top: "...comparing the profile log-likelihood of a parameter to the threshold for an approximate 95% confidence interval,..." Would it be simpler to say that identifiability is defined based on constructing approximate confidence intervals for each parameter?

p.15, "We produce points on the profile likelihood sequentially, using the log-likelihood at the MLE, the previous point, and the initially specified guess as an initial guess; ..." This is very unclear.

p.15, bottom and p.16, caption of Figure 5: It is not clear to me what "profiled error function" means. Also, it would be good to mention in the Figure caption that Figure 5e shows the noise-free case, in contrast to the other panels.

p.19: "Furthermore, this analysis demonstrates that, roughly speaking, the parameter subset (Q,Rd,γ) correspond primarily to the maximum spheroid size. In other words, there exists a trivariate function of (Q, Rd, γ) that maps to Rmax, and by extension the likelihood." - It is not clear to me how the authors come to this conclusion, here, or in similar cases below. I assume that it is related to implicit assumptions which components of the sensitivity matrix are "small" - I think this should be explained explicitly.

p.19, "... a direction in which the model relatively insensitive" -> a direction in which the model IS relatively insensitive

p.21, second line below subsection heading: "with respect to model complexity and identifiable" -> with respect to model complexity and IDENTIFIABILITY

p.22, above subsection heading: "In this case, we solve eq. (32) using the FIM for the radial-death model, and terminate the solution where the solution drops below the profile- likelihood-based threshold for a 95% confidence interval." - I have no idea what the authors want to say here.

p.23, "The reason for number of parameters in the W&K model is the biological granu- larity it provides:" - This sentence is unclear and might be incomplete. Do the authors explain why the W&K model has more parameters than other models investigated here?

p.25: "... we can use the rows sensitivity matrix, ..." - incomplete sentence

p.25: "... , mathematical models can always be made arbitrarily complex." - I think that GOOD modellers would not make models arbitrarily complex.

p.25, "... nutrient dependancies..." -> nutrient dependencies

p.25, bottom line: "... however not being constructionist, ..." - Maybe "constructivist"?

p.25-26, bottom to top: "... they limit the conversation between the mathematics and the biology that allows impactful insights to be made." - This sentence sounds a bit bombastic :-)

p.26: "... objectively studying the geometry of a map from the parameters in the complex model to those in the identifiable surrogate." - What does "objectively" mean in this case?

p.26: "One interpretation of our approach is of a layer that sits between the model parameters and the likelihood (or other goodness-of-fit metric) that is traditionally studied in identifiability and model sloppiness analysis." - I am not sure what the authors want to say here.

p.26: "... data from these experiments can be even more limited" - What do the authors want to say here? Does this mean that these data sets are rare?

p.26: "..., if the complex model is considered reality, the surrogate model produces predictions that are biased" - I am not sure what this means. Which parameters in the surrogate model are biased and why?

p.26: "... implicitly incorporates the noisy nature of the experimental measurements." - what does this mean?

p.27: "Recent work considers identifiability and sloppiness analysis based directly on the parameter covariance matrix estimated from Bayesian methods such as Markov-chain Monte-Carlo to provide an overall snapshot of the global parameter sensitivities. However, we expect this approach to be problematic in our geometric framework, since the model-map is based on an equivalence between models that may only apply locally." - I find it hard to follow this argument. Why do the authors think that Bayesian approaches might not be applicable and what do they mean when they say that equivalence between models "may only apply locally"?

p.28: "Building up a global model-map between the complex and surrogate models is another approach to overcome the localisation limitation of our methods, and could be computationally advantageous in the case of computationally expensive complex models." - What does "localisation limitation" mean?

p.28: "...complex models are numerous and conformable to the questions of interest" - What does "conformable to the question of interest" mean?</r_d?></r_d?

Reviewer #2: Geometric analysis enables biological insight from complex non-identifiable models using simple surrogates

Authors’ also provide compelling explanation of the limited utility of AIC when fitting mechanistic models to data (e.g. models with high AIC values may provide some utility if parameters of interest are identifiable). The authors then propose a method to study models with non-identifiable parameters using identifiable models w/ similar behavior. The models introduced in figure 1 have indistinguishable AIC, and model equivalence can be studied. The authors’ approach to begin with a model of intermediate complexity (Greenspan) then add noise and fit alternative models to this synthetic data, is a nice approach to understand the utility of their methods, with some objective ground truth known. The paper’s exposition is generally very clear and the results and methods are exciting & a good fit for publication at Plos CB. For example, the comparison between models (e.g. eqn 31) is a powerful and intuitive result from the methods proposed here.

Below I outline a few minor suggests to improve the clarity of the manuscript. These are all minor corrections to help clarify a few points that were initially confusing but clarified in later points of the manuscript.

1. Equation 2 implies that this method is used to fit the model to one output data only. Can this method be extended to multiple data streams collected simultaneously (e.g. tumor radius and nutrient level simulataneously)? I think some brief discussion of this should be mentioned in the intro and/or discussion.

2. Page 7, “…which is consistent with experiments where these quantities are not directly measured.” Perhaps it’s just the wording of this sentence, but I am confused about this sentence. Do the authors mean that every dependent variable that does not have a corresponding experimental value for comparison is non-dimensionalized?

3. Page 7, the phrase “which we have previously validated against experimental data” is repeated twice (first & fourth paragraphs) – probably only necessary to state once.

4. Page 14, second last paragraph: “…the MLE subject to the constrain” – I think this should be “constraint” not “constrain”.

5. Section 3.1.1 can be improved by clarifying the description of identifiability in figure 5. For example “we can establish identifiability by comparing the profile log-likelihood of a parameter to the threshold for an approximate 95% confidence interval” – it was not initially clear to me how the “comparing the profile” was done. Are we looking for ranges where PLL > 95% or PLL < 95%? Figure 5 is a great visual, supporting eqn. 22, but more brief explanation would help the reader. I also think it would be helpful to briefly note how identifiability would be altered w/ a new confidence interval (perhaps in the caption of figure 5).

6. After reading subsequent sections, I think the text on page 5, 1st paragraph, last sentence can be clarified: “… relatively large eigenvalues (informative directions) separated by several decades from one or two sloppy directions.” It wasn’t initially clear to me if 1) the separation or 2) the magnitude is the indicator of informative vs non-informative (sloppy) directions. I’m still unclear how this related to the claim on page 19 that the “dimensionality of this manifold corresponds to the number of sloppy directions” – with reference to figure 1d. From visual inspection of Figure 6, I’m not sure if I’m drawing the correction conclusion about the dimensionality of the manifold (as each subpanel appears to be a one-dimensional manifold).

Reviewer #3: This is an interesting and intriguing paper. Its main contribution is a method to map parameters (or combinations thereof) between different models. Thus, some kind of correspondence between the parameters in a complex, unidentifiable model and the parameters in a simple identifiable model can be established. This is useful because sometimes a parameter appearing in two different models under the same name is intuitively / conceptually thought to play the same role in both models (i.e. to have the exact same physical meaning), but this is not necessarily so. Importantly, the simpler (also called surrogate) model does not need to be nested within the complex model.

My main concern is about the organization of the article:

- While the paper is, generally speaking, well written, I found it too lengthy. I think it is difficult to grasp the precise nature of the main contribution, and it is easy to lose track of the main message during the reading. Perhaps the authors could consider moving some material to the Supplementary Information file. For example, Section 2.1 (Models) could be shortened by doing so, leaving only a very brief description of the models (maybe a Table would be useful) as well as the figures and tables.

- It is a bit weird that Section 2, “Methods”, has only one subsection, which describes models but not methods. Instead, identifiability methods are described in Section 3.1. I would suggest moving the description of those methods to a new subsection 2.2, and leaving 3.1 to the application of the methods to the models.

- As for Subsection 3.2, it presents the methodology with which the model-parameter map is obtained, which is the main contribution of the article. I think this should be highlighted more. Also, the methodology should be clearly defined, in the most general way possible, and with sufficient details. If I understood it correctly, the main result would be the construction of the sensitivity matrix (eqs. 29—30), which is obtained from the Jacobian in eq. (28). The calculation of said Jacobian is not trivial at all, and more details should be given about it.

I also have a number of more specific comments:

- Page 1: I am not fully convinced by this part of the last sentence in the Abstract: “Our geometric approach is able to (…) subset non-identifiable parameter combinations that relate to individual data features (…)”. It is unclear to me how parameter combinations relate to “individual data features”. If this sentence is kept, the authors should better explain it in the article.

- Page 4: “In comparison to model selection criterion like AIC, identifiability analysis provides an often subjective view of the identifiability of individual model parameters.” I find it a bit weird to use the word “subjective” here.

- Page 9, equation (4): is Rn(t) = 0? If yes, then subsequent equations would be simplified…

- Page 19: in the paragraph immediately before section 3.2.1 it is argued that the approach can be used to finding identifiable combinations of individually unidentifiable parameters. (Thanks to the map from a simple, identifiable model to a complex, unidentifiable one). The authors could mention that there are already other methods to achieve that goal, e.g. based on Profile Likelihood (https://doi.org/10.1371/journal.pone.0162366) or FIM-based sensitivity analysis (see https://doi.org/10.1016/j.automatica.2015.05.004 and more recent works by the first author).

- Page 19: “Furthermore, this analysis demonstrates that, roughly speaking, the parameter subset (Q; Rd; gamma) correspond primarily to the maximum spheroid size.” Why is it so? It is not obvious to me.

- Page 21: “Secondly, we see that gamma and lambda in the Greenspan model have a near one-to-one correspondence to zeta in the radial-death model.” This does not seem right; shouldn’t it be only gamma, not gamma and lambda?

- Page 22: “… directions (if any) in the parameter space to which the model is insensitive…” � it should probably be specified that what is insensitive is “the model output”, not simply “the model”.

- Page 27: “Recent work considers identifiability and sloppiness analysis based directly on the parameter covariance matrix estimated from Bayesian methods…”. Please provide references to that work.

- A few typos:

o In page 3, “One of many criterion used…” should be “criteria”.

o In page 17, “For example, find that the condition number…” should be “For example, we find…”, or something similar.

**Have the authors made all data and (if applicable) computational code underlying the findings in their manuscript fully available?**

Reviewer #1: Yes

Reviewer #2: Yes

Reviewer #3: Yes

PLOS authors have the option to publish the peer review history of their article (what does this mean?). If published, this will include your full peer review and any attached files.

Reviewer #1: **Yes: **Ivo Siekmann

Reviewer #2: No

Reviewer #3: **Yes: **Alejandro F. Villaverde

Figure Files:

Data Requirements:

Reproducibility:

References:

---

## [Decision Letter · Decision Letter 1]

26 Dec 2022

Dear Professor Simpson,

We are pleased to inform you that your manuscript 'Geometric analysis enables biological insight from complex non-identifiable models using simple surrogates' has been provisionally accepted for publication in PLOS Computational Biology.

Best regards,

Nicholas Mancuso

Guest Editor

PLOS Computational Biology

Douglas Lauffenburger

Section Editor

PLOS Computational Biology

Reviewer's Responses to Questions

**Comments to the Authors:**

Reviewer #1: I am commenting on a revision of the manuscript "Geometric analysis enables biological insight from complex non-identifiable models using simple surrogates" by Browning and Simpson. The authors have submitted an excellent revision and all my comments have been addressed. In my opinion this is very nice work and I am happy to recommend the paper to be accepted.

Regarding the logistic model I thank the authors for their response and I would like to make an additional remark. The most general "mechanistic" interpretation of the logistic growth model is "intraspecific competition". This includes space but also multiple other factors including nutrients or oxygen that the authors mention in their response. Of course, the model is very crude, a mass action term with a negative sign, but a mechanistic interpretation is that whenever individuals of a population meet they compete with each other for the resources that can be found in a particular habitat. I agree that it is debatable if this should indeed be considered "mechanistic" because, as the authors explain, the underlying biophysical mechanisms of intraspecific competition are not explicitly accounted for by the model. From that point of view I completely agree with the authors that the model is phenomenological... as usual it is a matter of perspective! :-)

Reviewer #2: Thank you to the authors for addressing my previous minor concerns. Very nice work!

Reviewer #3: The authors have addressed all my previous comments. I have no further suggestions.

**Have the authors made all data and (if applicable) computational code underlying the findings in their manuscript fully available?**

Reviewer #1: Yes

Reviewer #2: None

Reviewer #3: Yes

PLOS authors have the option to publish the peer review history of their article (what does this mean?). If published, this will include your full peer review and any attached files.

Reviewer #1: **Yes: **Ivo Siekmann

Reviewer #2: **Yes: **Jeffrey West

Reviewer #3: **Yes: **Alejandro F. Villaverde

---

## [Editor Report · Acceptance letter]

9 Jan 2023

PCOMPBIOL-D-22-01206R1 

Geometric analysis enables biological insight from complex non-identifiable models using simple surrogates

Dear Dr Simpson,

I am pleased to inform you that your manuscript has been formally accepted for publication in PLOS Computational Biology. Your manuscript is now with our production department and you will be notified of the publication date in due course.

With kind regards,

Anita Estes
